# HIGH PROBABILITY BOUNDS FOR A CLASS OF NON-CONVEX ALGORITHMS WITH ADAGRAD STEPSIZE

**Ali Kavis**
EPFL (LIONS)
`ali.kavis`
`@epfl.ch`

**Kfir Y. Levy** [*]
Technion
`kfirylevy`
`@technion.ac.il`

**Volkan Cevher**
EPFL (LIONS)
`volkan.cevher`
`@epfl.ch`

## ABSTRACT

In this paper, we propose a new, simplified high probability analysis of AdaGrad for smooth, non-convex problems. More specifically, we focus on a particular accelerated gradient (AGD) template (Lan, 2020), through which we recover the original AdaGrad and its variant with averaging, and prove a convergence rate of $\mathcal{O}(1/\sqrt{T})$ with high probability without the knowledge of smoothness and variance. We use a particular version of Freedman's concentration bound for martingale difference sequences (Kakade & Tewari, 2008) which enables us to achieve the best-known dependence of $\log(1/\delta)$ on the probability margin $\delta$. We present our analysis in a modular way and obtain a complementary $\mathcal{O}(1/T)$ convergence rate in the deterministic setting. To the best of our knowledge, this is the first high probability result for AdaGrad with a truly adaptive scheme, i.e., completely oblivious to the knowledge of smoothness and uniform variance bound, which simultaneously has best-known dependence of $\log(1/\delta)$. We further prove noise adaptation property of AdaGrad under additional noise assumptions.

## 1 INTRODUCTION

Adaptive gradient methods are a staple of machine learning (ML) in solving core problems such as

$$\min_{x \in \mathbb{R}^d} f(x) := \mathbb{E}_{z \sim \mathcal{D}}\left[f(x; z)\right], \tag{P}$$

where the objective $f(x)$ is possibly non-convex, and $\mathcal{D}$ is a probability distribution from which the random vector $z$ is drawn. Problem (P) captures for instance empirical risk minimization or finite-sum minimization (Shalev-Shwartz & Ben-David, 2014) problems, where $z$ represents the mini-batches and $\mathcal{D}$ corresponds to the distribution governing the data or its sampling strategy.

Within the context of large-scale problems, including streaming data, computing full gradients is extremely costly, if not impossible. Hence, stochastic iterative methods are the main optimizer choice in these scenarios. The so-called adaptive methods such as AdaGrad (Duchi et al., 2011), Adam (Kingma & Ba, 2014) and AmsGrad (Reddi et al., 2018) have witnessed a surge of interest both theoretically and practically due to their off-the-shelf performance. For instance, adaptive optimization methods are known to show superior performance in various learning tasks such as machine translation (Zhang et al., 2020; Vaswani et al., 2017).

From a theoretical point of view, existing literature provides a quite comprehensive understanding regarding the *expected* behaviour of existing adaptive learning methods. Nevertheless, these results do not capture the behaviour of adaptive methods within a single/few runs, which is related to the *probabilistic* nature of these methods. While there exists *high probability* analysis of vanilla SGD for non-convex problems (Ghadimi & Lan, 2013), adaptive methods have received limited attention in this context.

Our main goal in this paper is to understand the probabilistic convergence properties of adaptive algorithms, specifically AdaGrad, while focusing on their *problem parameter* adaptation capabilities in the non-convex setting. In this manuscript, adaptivity refers to the ability of an algorithm to ensure

---

[*]A Viterbi fellow

convergence without requiring the knowledge of quantities such as smoothness modulus or variance of noise. Studies along this direction largely exist for the convex objectives; for instance, Levy et al. (2018) shows that AdaGrad can (implicitly) exploit smoothness and adapt to the magnitude of noise in the gradients when $f(x)$ is convex in (P).

This alternative perspective to adaptivity is crucial because most existing analysis, both for classical and adaptive methods, assume to have access to smoothness constant, bound on gradients (Reddi et al., 2018) and even noise variance (Ghadimi & Lan, 2013). In practice, it is difficult, if not impossible, to compute or even estimate such quantities. For this purpose, in the setting of (P) we study a class of adaptive gradient methods that enable us to handle noisy gradient feedback without requiring the knowledge of the objective's smoothness modulus, noise variance or a bound on gradient norms.

We summarize our contributions as follows:

1. We provide a modular, simple high probability analysis for AdaGrad-type adaptive methods.
2. We present the first *optimal high probability convergence result* of the *original AdaGrad* algorithm for non-convex smooth problems. Concretely,
   (a) we analyze a fully adaptive step-size, oblivious to Lipschitz constant and noise variance,
   (b) we obtain the best known dependence of $\log(1/\delta)$ on the probability margin $\delta$.
   (c) we show that under sub-Gaussian noise model, AdaGrad adapts to noise level with high probability, i.e, as variance $\sigma \to 0$, convergence rate improves, $1/\sqrt{T} \to 1/T$.
3. We present new extensions of AdaGrad that include averaging and momentum primitives, and prove similar high probability bounds for these methods, as well. Concretely, we study a general adaptive template which individually recovers AdaGrad, AdaGrad with averaging and adaptive RSAG (Ghadimi & Lan, 2016) for different parameter choices.

In the next section, we will provide a broad overview of related work with an emphasis on the recent developments. Section 3 formalizes the problem setting and states our blanket assumptions. Section 4 introduces the building blocks of our proposed proof technique while proving convergence results for AdaGrad. We generalize the convergence results of AdaGrad for a class of nonconvex, adaptive algorithms in Section 5. Finally, we present concluding remarks in the last section.

## 2 RELATED WORK

**Adaptive methods for stochastic optimization** As an extended version of the online (projected) GD (Zinkevich, 2003), AdaGrad (Duchi et al., 2011) is the pioneering work behind most of the contemporary adaptive optimization algorithms Adam, AmsGrad and RmsProp (Tieleman & Hinton, 2012) to name a few. Simply put, such AdaGrad-type methods compute step-sizes on-the-fly by accumulating gradient information and achieve adaptive regret bounds as a function of gradient history (see also (Tran & Phong, 2019; Alacaoglu et al., 2020b; Luo et al., 2019; Huang et al., 2019)).

**Universality, adaptive methods and acceleration** We call an algorithm *universal* if it achieves optimal rates under different settings, without any modifications. For *convex* minimization problems, Levy et al. (2018) showed that AdaGrad attains a rate of $\mathcal{O}(1/T + \sigma/\sqrt{T})$ by implicitly adapting to smoothness and noise levels; here $T$ is the number of oracle queries and $\sigma$ is the noise variance. They also proposed an accelerated AdaGrad variant with scalar step-size. The latter result was extended for compactly constrained problems via accelerated Mirror-Prox algorithm (Kavis et al., 2019), and for composite objectives (Joulani et al., 2020). Recently, Ene et al. (2021) have further generalized the latter results by designing a novel adaptive, accelerated algorithm with per-coordinate step-sizes. Convergence properties of such algorithms under smooth, non-convex losses are unknown to date.

**Adaptive methods for nonconvex optimization** Following the popularity of neural networks, adaptive methods have attracted massive attention due to their favorable performance in training and their ease of tuning. The literature is quite vast, which is impossible to cover exhaustively here. Within the representative subset (Chen et al., 2019; Zaheer et al., 2018; Li & Orabona, 2019; Zou et al., 2019; Defossez et al., 2020; Alacaoglu et al., 2020a; Chen et al., 2020; Levy et al., 2021). The majority of the existing results on adaptive methods for nonconvex problems focus on *in expectation* performance.

**High probability results** Ghadimi & Lan (2013) are the first to analyze SGD in the non-convex regime and provide tight convergence bounds. Nevertheless, their method requires a prior knowledge of the smoothness modulus and noise variance. In the context of adaptive methods, Li & Orabona (2020) considers delayed AdaGrad (with lag-one-behind step-size) for smooth, non-convex losses under subgaussian noise and proved $O(\sigma\sqrt{\log(T/\delta)}/\sqrt{T})$ rate. Under similar conditions, Zhou et al. (2018) proves convergence of order $O((\sigma^2\log(1/\delta))/T + 1/\sqrt{T})$ for AdaGrad. However, both works require the knowledge of smoothness to set the step-size. Moreover, Ward et al. (2019) guarantees that AdaGrad with scalar step-size convergences at $O((1/\delta)\log(T)/\sqrt{T})$ rate with high probability. Although their framework is oblivious to smoothness constant, their dependence of probability margin is far from optimal. More recently, under heavy-tailed noise having bounded p[th] moment for $p \in (1, 2)$, Cutkosky & Mehta (2021) proves a rate of $O(\log(T/\delta)/T^{(p-1)/(3p-2)})$ for clipped normalized SGD with momentum; nevertheless their method requires the knowledge of (a bound on) the behavior of the heavy tails.

## 3 SETUP AND PRELIMINARIES

As we stated in the introduction, we consider the unconstrained minimization setting

$$\min_{x\in\mathbb{R}^d} f(x) := \mathbb{E}_{z\sim\mathcal{D}}\left[f(x; z)\right],$$

where the differentiable function $f : \mathbb{R}^d \to \mathbb{R}$ is a smooth and (possibly) nonconvex function.

We are interested in finding a first-order $\epsilon$-stationary point satisfying $\|\nabla f(x_t)\|^2 \le \epsilon$, where $\|\cdot\|$ denotes the Euclidean norm for the sake of simplicity. As the standard measure of convergence, we will quantify the performance of algorithms with respect to *average* gradient norm, $\frac{1}{T}\sum_{t=1}^{T}\|\nabla f(x_t)\|^2$. It immediately implies convergence in *minimum* gradient norm, $\min_{t\in[T]}\|\nabla f(x_t)\|^2$. Moreover, note that if we are able to bound $\frac{1}{T}\sum_{t=1}^{T}\|\nabla f(x_t)\|^2$, then by choosing to output a solution $\bar{x}_T$ which is chosen uniformly at random from the set of query points $\{x_1,\ldots,x_T\}$, then we ensure that $\mathbf{E}\|\nabla f(\bar{x}_T)\|^2 := \frac{1}{T}\sum_{t=1}^{T}\|\nabla f(x_t)\|^2$ is bounded.

A function is called $G$-Lipschitz continuous if it satisfies

$$|f(x) - f(y)| \le G\|x - y\|, \quad \forall x, y \in \text{dom}(f), \tag{1}$$

which immediately implies that

$$\|\nabla f(x)\| \le G, \quad \forall x \in \text{dom}(f). \tag{2}$$

A differentiable function is called $L$-smooth if it has $L$-Lipschitz gradient

$$\|\nabla f(x) - \nabla f(y)\| \le L\|x - y\|, \quad \forall x, y \in \text{dom}(\nabla f). \tag{3}$$

An equivalent characterization is referred to as the "descent lemma" (Ward et al., 2019; Beck, 2017),

$$|f(x) - f(y) - \langle\nabla f(y), x - y\rangle| \le (L/2)\|x - y\|^2. \tag{4}$$

**Assumptions on oracle model:** We denote stochastic gradients with $\widetilde{\nabla}f(x) = \nabla f(x; z)$, for some random vector drawn from distribution $\mathcal{D}$. Since our template embraces single-call algorithms, we use this shorthand notation for simplicity. An oracle is called (conditionally) unbiased if

$$\mathbb{E}\left[\widetilde{\nabla}f(x)|x\right] = \nabla f(x), \quad \forall x \in \text{dom}(\nabla f). \tag{5}$$

Gradient estimates generated by a first-order oracle are said to have bounded variance if they satisfy

$$\mathbb{E}\left[\|\widetilde{\nabla}f(x) - \nabla f(x)\|^2|x\right] \le \sigma^2, \quad \forall \in \text{dom}(\nabla f). \tag{6}$$

Finally, we assume that the stochastic gradient are bounded almost surely, i.e.,

$$\|\widetilde{\nabla}f(x)\| \le \tilde{G}, \quad \forall x \in \text{dom}(\nabla f). \tag{7}$$

**Remark 1.** *Bounded variance assumption* (6) *is standard in the analysis of stochastic methods (Lan, 2020). Similarly, for the analysis of adaptive methods in the nonconvex realm, it is very common to assume bounded stochastic gradients (see (Zaheer et al., 2018; Zhou et al., 2018; Chen et al., 2019; Li & Orabona, 2020) and references therein).*

## 4  PROPOSED ANALYSIS AND ADAGRAD

This section introduces our proposed proof technique as well as our main theoretical results for AdaGrad with proof sketches and discussions on the key elements of our theoretical findings. We will present a high-level overview of our simplified, modular proof strategy while proving a complementary convergence result for AdaGrad under deterministic oracles. In the sequel, we refer to the name AdaGrad as the scalar step-size version (also known as AdaGrad-Norm) as presented in Algorithm 1.

---

**Algorithm 1** AdaGrad

---

**Input:** time horizon $T$, $x_1 \in \mathbb{R}^d$, step-size $\{\eta_t\}_{t \in [T]}$, $G_0 > 0$
 1: **for** $t = 1, ..., T$ **do**
 2:     Generate $g_t = \widetilde{\nabla} f(x_t)$
 3:     $\eta_t = \frac{1}{\sqrt{G_0^2 + \sum_{k=1}^{t} \|g_k\|^2}}$
 4:     $x_{t+1} = x_t - \eta_t g_t$
 5: **end for**

---

Before moving forward with analysis, let us first establish the notation we will use to simplify the presentation. In the sequel, we use $[T]$ as a shorthand expression for the set $\{1, 2, ..., T\}$. We will use $\Delta_t = f(x_t) - \min_{x \in \mathbb{R}^d} f(x)$ as a concise notation for objective suboptimality and $\Delta_{\max} = \max_{t \in [T+1]} \Delta_t$ will denote the maximum over $\Delta_t$. In the rest of Section 4, we denote stochastic gradient of $f$ at $x_t$ by $g_t = \widetilde{\nabla} f(x_t) = \nabla f(x_t; z_t)$ and its deterministic equivalent as $\bar{g}_t := \nabla f(x_t)$. We also use the following notation for the noise vector $\xi_t := g_t - \bar{g}_t$.

Notice that AdaGrad (Alg. 1) does not require any prior knowledge regarding the smoothness modulus nor the noise variance. The main results in this section is Theorem 4.2, where we show that with high probability AdaGrad obtains an optimal rate $\tilde{O}(\log(1/\delta)/\sqrt{T})$ for finding an approximate stationary point. Moreover, Theorem 4.1 shows that in the deterministic case AdaGrad achieves an optimal rate of $O(1/T)$, thus establishing its universality.

### 4.1  TECHNICAL LEMMAS

We make use of a few technical lemmas while proving our main results, which we refer to in our proof sketches. We present them all at once before the main theorems for completeness. First, Lemmas 4.1 and 4.2 are well-known results from online learning, essential for handling adaptive stepsizes.

**Lemma 4.1.** *Let $a_1, ..., a_n$ be a sequence of non-negative real numbers. Then, it holds that*

$$\sqrt{\sum_{i=1}^{n} a_i} \leq \sum_{i=1}^{n} \frac{a_i}{\sqrt{\sum_{k=1}^{i} a_k}} \leq 2\sqrt{\sum_{i=1}^{n} a_i}$$

**Lemma 4.2.** *Let $a_1, ..., a_n$ be a sequence of non-negative real numbers. Then, it holds that*

$$\sum_{i=1}^{n} \frac{a_i}{\sum_{k=1}^{i} a_i} \leq 1 + \log\left(1 + \sum_{i=1}^{n} a_i\right)$$

The next lemma is the key for achieving the high probability bounds.

**Lemma 4.3** (Lemma 3 in Kakade & Tewari (2008))**.** *Let $X_t$ be a martingale difference sequence such that $|X_t| \leq b$. Let us also define*

$$\mathbf{Var}_{t-1}(X_t) = \mathbf{Var}\left(X_t \mid X_1, ..., X_{t-1}\right) = \mathbb{E}\left[X_t^2 \mid X_1, ..., X_{t-1}\right],$$

*and $V_T = \sum_{t=1}^{T} \mathbf{Var}_{t-1}(X_t)$ as the sum of variances. For $\delta < 1/e$ and $T \geq 3$, it holds that*

$$\mathbb{P}\left(\sum_{t=1}^{T} X_t > \max\left\{2\sqrt{V_T}, 3b\sqrt{\log(1/\delta)}\right\}\sqrt{\log(1/\delta)}\right) \leq 4\log(T)\delta \tag{8}$$

### 4.2 OVERVIEW OF PROPOSED ANALYSIS

We will start by presenting the individual steps of our proof and provide insight into its advantages. In the rest of this section, we solely focus on AdaGrad, however, the same intuition applies to the more general Algorithm 2 as we will make clear in the sequel. Using the shorthand notations of $\bar{g}_t = \nabla f(x_t)$ and $g_t = \nabla f(x_t; z_t) = \bar{g}_t + \xi_t$, the classical analysis begins with,

$$f(x_{t+1}) - f(x_t) \leq -\eta_t \|\bar{g}_t\|^2 - \eta_t \langle \bar{g}_t, \xi_t \rangle + \frac{L\eta_t^2}{2}\|g_t\|^2.$$

which is due to the smoothness property in Eq. (4). Re-arranging and summing over $t \in [T]$ yields,

$$\sum_{t=1}^{T} \eta_t \|\bar{g}_t\|^2 \leq f(x_1) - f(x^*) + \sum_{t=1}^{T} -\eta_t \langle \bar{g}_t, \xi_t \rangle + \frac{L}{2}\sum_{t=1}^{T} \eta_t^2 \|g_t\|^2.$$

The main issue in this expression is the $\eta_t \langle \bar{g}_t, \xi_t \rangle$ term, which creates measurability problems due to the fact hat $\eta_t$ and $\xi_t$ are dependent random variables. On the left hand side, the mismatch between $\eta_t$ and $\|\bar{g}_t\|^2$ prohibits the use of technicals lemmas as we accumulate *stochastic* gradients for $\eta_t$. Moreover, we cannot make use of Holder-type inequalities as we deal with high probability results. Instead, we divide both sides by $\eta_t$, then sum over $t$ and re-arrange to obtain a bound of the form,

$$\sum_{t=1}^{T} \|\bar{g}_t\|^2 \leq \frac{\Delta_1}{\eta_1} + \sum_{t=2}^{T} \left( \frac{1}{\eta_t} - \frac{1}{\eta_{t-1}} \right) \Delta_t + \sum_{t=1}^{T} -\langle \bar{g}_t, \xi_t \rangle + \frac{L}{2}\sum_{t=1}^{T} \eta_t \|g_t\|^2$$

$$\leq \frac{\Delta_{\max}}{\eta_T} + \sum_{t=1}^{T} -\langle \bar{g}_t, \xi_t \rangle + \frac{L}{2}\sum_{t=1}^{T} \eta_t \|g_t\|^2 \qquad (9)$$

This modification solves the two aforementioned problems, but we now need to ensure boundedness of function values, specifically the maximum distance to the optimum, $\Delta_{\max}$. In fact, neural networks with bounded activations (e.g. sigmoid function) in the last layer and some objective functions in robust non-convex optimization (e.g. Welsch loss (Barron, 2019)) satisfy bounded function values. However, this is a restrictive assumption to make for general smooth problems and we will *prove* that it is bounded or at least it grows no faster than $O(\log(T))$. As a key element of our approach, we show that it is the case for Alg. 1 & 2. Now, we are at a position to state an overview of our proof:

1. Show that $\Delta_{\max} \leq O(\log(T))$ with high probability or $\Delta_{\max} \leq O(1)$ (deterministic).
2. Prove that $\sum_{t=1}^{T} -\langle \bar{g}_t, \xi_t \rangle \leq \tilde{O}(\sqrt{T})$ with high probability using Lemma 4.3.
3. Show that $\frac{L}{2}\sum_{t=1}^{T} \eta_t \|g_t\|^2 \leq O(\sqrt{T})$ by using Lemma 4.1.

For completeness, we will propose a simple proof for AdaGrad in the deterministic setting. This will showcase advantages of our approach, while providing some insight into it. We provide a sketch of the proof, whose full version will be accessible in the appendix.

**Theorem 4.1.** *Let $x_t$ be generated by Algorithm 1 with $G_0 = 0$ for simplicity. Then, it holds that*

$$\frac{1}{T}\sum_{t=1}^{T} \|\nabla f(x_t)\|^2 \leq O\left( \frac{(\Delta_1 + L)^2}{T} \right).$$

*Proof Sketch (Theorem 4.1).* Setting $\bar{g}_t = \nabla f(x_t)$, deterministic counterpart of Eq. (9) becomes

$$\sum_{t=1}^{T} \|\bar{g}_t\|^2 \;\leq\; \frac{\Delta_{\max}}{\eta_T} + \frac{L}{2}\sum_{t=1}^{T} \eta_t \|\bar{g}_t\|^2 \;\leq\; (\Delta_{\max} + L)\sqrt{\sum_{t=1}^{T} \|\bar{g}_t\|^2},$$

where we obtain the final inequality using Lemma 4.1. Now, we show that $\Delta_{T+1}$ is bounded for any $T$. Using descent lemma and summing over $t \in [T]$,

$$f(x_{T+1}) - f(x^*) \leq f(x_1) - f(x^*) + \sum_{t=1}^{T} \left( \frac{L\eta_t}{2} - 1 \right) \eta_t \|\bar{g}_t\|^2.$$

Now, define $t_0 = \max\left\{t \in [T] \mid \eta_t > \frac{2}{L}\right\}$, such that $\left(\frac{L\eta_t}{2} - 1\right) \leq 0$ for any $t > t_0$. Then,

$$f(x_{T+1}) - f(x^*) \leq \Delta_1 + \sum_{t=1}^{t_0} \left(\frac{L\eta_t}{2} - 1\right) \eta_t \|\bar{g}_t\|^2 + \sum_{t=t_0+1}^{T} \left(\frac{L\eta_t}{2} - 1\right) \eta_t \|\bar{g}_t\|^2$$

$$\leq \Delta_1 + \frac{L}{2} \sum_{t=1}^{t_0} \eta_t^2 \|\bar{g}_t\|^2 \leq \Delta_1 + \frac{L}{2}\left(1 + \log\left(1 + L^2/4\right)\right),$$

where we use the definition of $t_0$ and Lemma 4.2 for the last inequality. Since this is true for any $T$, the bound holds for $\Delta_{\max}$, as well. Defining $X = \sqrt{\sum_{t=1}^{T} \|\bar{g}_t\|^2}$, the original expression reduces to $X^2 \leq (\Delta_{\max} + L) X$. Solving for $X$, plugging in the bound for $\Delta_{\max}$ and dividing by $T$ results in

$$\frac{1}{T} \sum_{t=1}^{T} \|\nabla f(x_t)\|^2 \leq \frac{\left(\Delta_1 + \frac{L}{2}\left(3 + \log\left(L^2/4\right)\right)\right)^2}{T}.$$

$\square$

**Remark 2.** *To our knowledge, the most relevant analysis was provided by Ward et al. (2019), which achieves $\mathcal{O}(\log(T)/T)$ convergence rate. Our new approach enables us to remove $\log(T)$ factor.*

### 4.3 HIGH PROBABILITY CONVERGENCE UNDER STOCHASTIC ORACLE

Having introduced the building blocks, we will now present the high probability convergence bound for AdaGrad (Algorithm 1). Let us begin by the departure point of our proof, which is Eq. (9)

$$\sum_{t=1}^{T} \|\bar{g}_t\|^2 \leq \underbrace{\frac{\Delta_{\max}}{\eta_T}}_{(*)} + \underbrace{\sum_{t=1}^{T} -\langle \bar{g}_t, \xi_t \rangle}_{(**)} + \underbrace{\frac{L}{2} \sum_{t=1}^{T} \eta_t \|g_t\|^2}_{(***)}. \tag{10}$$

We can readily bound expression $(***)$ using Lemma 4.1. Hence, what remains is to argue about high probability bounds for expressions $(*)$ and $(**)$, which we do in the following propositions.

**Proposition 4.1.** *Using Lemma 4.3, with probability $1 - 4\log(T)\delta$ and $\delta < 1/e$, we have*

$$\sum_{t=1}^{T} -\langle \bar{g}_t, \xi_t \rangle \leq 2\sigma\sqrt{\log(1/\delta)}\sqrt{\sum_{t=1}^{T} \|\bar{g}_t\|^2} + 3(G^2 + G\tilde{G})\log(1/\delta).$$

The last ingredient of the analysis is the bound on $\Delta_t$. The following proposition ensures a high probability bound of order $O(\log(t))$ on $\Delta_t$ under Algorithm 1.

**Proposition 4.2.** *Let $x_t$ be generated by AdaGrad for $G_0 > 0$. With probability at least $1 - 4\log(t)\delta$,*

$$\Delta_{t+1} \leq \Delta_1 + 2L\left(1 + \log\left(\max\{1, G_0^2\} + \tilde{G}^2 t\right)\right) + G_0^{-1}(M_1 + \sigma^2)\log(1/\delta) + M_2,$$

*where $M_1 = 3(G^2 + G\tilde{G})$ and $M_2 = G_0^{-1}(2G^2 + G\tilde{G})$.*

As an immediate corollary, since the statement of Proposition 4.2 holds for any $t$, it holds for $\Delta_{\max}$ by definition. Hence, we have that $\max_{t \in [T]} \Delta_t = \Delta_{\max} \leq O\left(\Delta_1 + L\log(T) + \sigma^2\log(1/\delta)\right)$ with high probability for any time horizon $T$. In the light of the above results, we are now able to present our high probability bound for AdaGrad.

**Theorem 4.2.** *Let $x_t$ be the sequence of iterates generated by AdaGrad. Under Assumptions 2, 6, 7, for $\Delta_{max} \leq O\left(\Delta_1 + L\log(T) + \sigma^2\log(1/\delta)\right)$, with probability at least $1 - 8\log(T)\delta$,*

$$\frac{1}{T} \sum_{t=1}^{T} \|\bar{g}_t\|^2 \leq \frac{(\Delta_{max} + L)G_0 + 3(G^2 + G\tilde{G})\log(1/\delta)}{T} + \frac{(\Delta_{max} + L)\tilde{G} + 2G\sigma\sqrt{\log(1/\delta)}}{\sqrt{T}}.$$

*Proof Sketch (Theorem 4.2).* Define $\bar{g}_t = \nabla f(x_t)$ and $g_t = \nabla f(x_t; z_t) = \bar{g}_t + \xi_t$. By Eq. (10),

$$\sum_{t=1}^{T} \|\bar{g}_t\|^2 \leq \frac{\Delta_{\max}}{\eta_T} + \sum_{t=1}^{T} -\langle \bar{g}_t, \xi_t \rangle + \frac{L}{2} \sum_{t=1}^{T} \eta_t \|g_t\|^2.$$

Invoking Lemma 4.1 on the last sum and using Proposition 4.1 for the second expression, we have with probability at least $1 - 4\log(T)\delta$

$$\sum_{t=1}^{T} \|\bar{g}_t\|^2 \leq (\Delta_{\max} + L) \sqrt{G_0^2 + \sum_{t=1}^{T} \|g_t\|^2} + 2\sigma \sqrt{\log(1/\delta)} \sqrt{\sum_{t=1}^{T} \|\bar{g}_t\|^2} + 3(G^2 + G\tilde{G})\log(1/\delta)$$

Finally, we use the bounds on the gradient norms $\|\bar{g}_t\| \leq G$ and $\|g_t\| \leq \tilde{G}$, re-arrange the terms and divide both sides by $T$. Due to Proposition 4.2, with probability at least $1 - 8\log(T)\delta$,

$$\frac{1}{T} \sum_{t=1}^{T} \|\bar{g}_t\|^2 \leq \frac{(\Delta_{\max} + L) G_0 + 3(G^2 + G\tilde{G})\log(1/\delta)}{T} + \frac{(\Delta_{\max} + L) \tilde{G} + 2G\sigma \sqrt{\log(1/\delta)}}{\sqrt{T}},$$

where $\Delta_{\max} \leq O\left(\Delta_1 + L\log(T) + \sigma^2 \log(\frac{1}{\delta})\right)$. We keep $\Delta_{\max}$ in the bound due to lack of space. $\quad\square$

## 4.4 Noise adaptation under sub-Gaussian noise model

To our knowledge, under the standard setting we consider (unbiased stochastic gradients with bounded variance), noise adaptation is not achieved for high probability convergence to first-order stationary points, specifically for AdaGrad-type adaptive methods. We call an algorithm *noise adaptive* if the convergence rate improves $1/\sqrt{T} \to 1/T$ as variance $\sigma \to 0$. Following the technical results and approach proposed by Li & Orabona (2020), we will prove that high probability convergence of AdaGrad (Algorithm 1) exhibits adaptation to noise under sub-Gaussian noise model. First, we will introduce the additional assumption on the noise. We assume that the tails of the noise behaves as sub-Gaussian if,

$$\mathbb{E}\left[\exp(\|\nabla f(x, z) - \nabla f(x)\|^2) \mid x\right] \leq \exp(\sigma^2). \tag{11}$$

This last assumption on the noise is more restrictive than standard assumption of bounded variance. Indeed, Eq. (11) implies bounded variance (Eq. (6)). Finally, we conclude with the main theorems. We first establish a high probability bound on $\Delta_{\max}$ and then present noise-adaptive rates for AdaGrad.

**Theorem 4.3.** *Let $x_t$ be generated by AdaGrad and define $\Delta_t = f(x_t) - \min_{x \in \mathbb{R}^d} f(x)$. Under sub-Gaussian noise assumption as in Eq. (11), with probability at least $1 - 3\delta$,*

$$\Delta_{t+1} \leq \Delta_1 + 3G_0^{-1}G^2 + 2G_0^{-1}\sigma^2 \log\left(\frac{et}{\delta}\right) + \frac{3}{4G_0}\sigma^2 \log(1/\delta)$$
$$+ \frac{L}{2}\left(1 + \log\left(\max\{1, G_0^2\} + 2G^2t + 2\sigma^2 t \log\left(\frac{et}{\delta}\right)\right)\right).$$

**Theorem 4.4.** *Let $x_t$ be generated by AdaGrad and define $\Delta_t = f(x_t) - \min_{x \in \mathbb{R}^d} f(x)$. Under sub-Gaussian noise assumption as in Eq. (11) and considering high probability boundedness of $\Delta_{max}$ due to Theorem 4.3, with probability at least $1 - 5\delta$,*

$$\frac{1}{T} \sum_{t=1}^{T} \|\bar{g}_t\|^2 \leq \frac{32\left(\Delta_{max} + L\right)^2 + 8\left(\Delta_{max} + L\right)\left(G_0 + \sigma\sqrt{2\log(1/\delta)}\right) + 8\sigma^2 \log(1/\delta)}{T} + \frac{8\sqrt{2}\left(\Delta_{max} + L\right)\sigma}{\sqrt{T}}.$$

**Remark 3.** *By introducing the sub-Gaussian noise model, we manage to achieve a high probability convergence bound that is* adaptive to noise, *while* removing the dependence on a bound on stochastic gradients *in the final result.*

## 5 Generalization to AGD template

Having proven the high probability convergence for AdaGrad, we will now present an extension of our analysis to the more general accelerated gradient (AGD) template, which corresponds to a specific reformulation of Nesterov's acceleration (Ghadimi & Lan, 2016).

---

**Algorithm 2** Generic AGD Template

---

**Input:** Horizon $T$, $\tilde{x}_1 = x_1 \in \mathbb{R}^d$, $\alpha_t \in (0, 1]$, step-sizes $\{\eta_t\}_{t \in [T]}$, $\{\gamma_t\}_{t \in [T]}$

1: **for** $t = 1, ..., T$ **do**
2: $\qquad \bar{x}_t = \alpha_t x_t + (1 - \alpha_t)\tilde{x}_t$
3: $\qquad$ Set $g_t = \nabla f(\bar{x}_t)$ or $g_t = \widetilde{\nabla} f(\bar{x}_t) = \nabla f(x_t; z_t)$
4: $\qquad x_{t+1} = x_t - \eta_t g_t$
5: $\qquad \tilde{x}_{t+1} = \bar{x}_t - \gamma_t g_t$
6: **end for**

---

This particular reformulation was recently referred to as linear coupling (Allen-Zhu & Orecchia, 2016), which is an intricate combination of mirror descent (MD), gradient descent (GD) and averaging. In the sequel, we focus on two aspects of the algorithm; averaging parameter $\alpha_t$ and selection of (adaptive) step-sizes $\eta_t$ and $\gamma_t$. We could recover some well-known algorithms from this generic scheme depending on parameter choices, which we display in Table 1.

Our reason behind choosing this generic algorithm is two-fold. First, it helps us demonstrate flexibility of our simple, modular proof technique by extending it to a generalized algorithmic template. Second, as an integral element of this scheme, we want to study the notion of averaging (equivalently momentum (Defazio, 2021)), which is an important primitive for machine learning and optimization problems. For instance, it is necessary for achieving *accelerated* convergence in convex optimization (Nesterov, 1983; Kavis et al., 2019), while it helps improve performance in neural network training and stabilizes the effect of noise (Defazio, 2021; Sutskever et al., 2013; Liu et al., 2020).

Next, we will briefly introduce instances of Algorithm 2, their properties and the corresponding parameter choices. We identify algorithms regarding the choice of averaging parameter $\alpha_t$ and step sizes $\eta_t$ and $\gamma_t$. The averaging parameter $\alpha_t$ has two possible forms: $\alpha_t = 2/(t+1)$ for weighted averaging and $\alpha_t = 1/t$ for uniform averaging. We take $\alpha_t = 2/(t+1)$ by default in our analysis, as it is a key element in achieving acceleration in the convex setting. Let us define the AdaGrad step size once more, which we use to define $\eta_t$ and $\gamma_t$,

$$\widetilde{\eta}_t = (G_0^2 + \sum_{k=1}^t \|g_k\|^2)^{-1/2}, \qquad G_0 > 0. \tag{12}$$

The first instance of Algorithm 2 is the AdaGrad, i.e. $x_{t+1} = x_t - \widetilde{\eta}_t g_t$. Since $\tilde{x}_1 = x_1$ by initialization, we have $\bar{x}_1 = \tilde{x}_1 = x_1$. The fact that $\eta_t = \gamma_t = \widetilde{\eta}_t$ implies the equivalence $\tilde{x}_t = x_t$ for any $t \in [T]$, which ignores averaging step. The second instance we obtain is AdaGrad with averaging,

$$\bar{x}_t = \alpha_t x_t + (1 - \alpha_t)\bar{x}_{t-1}$$
$$x_{t+1} = x_t - \alpha_t \eta_t g_t, \tag{13}$$

where $\eta_t = \widetilde{\eta}_t$ and $\gamma_t = 0$. For the initialization $x_1 = \tilde{x}_1$, we can obtain by induction that $\tilde{x}_t = \bar{x}_{t-1}$, hence the scheme above. The final scheme we will analyze is RSAG algorithm proposed by Ghadimi & Lan (2016). It selects a step size pair that satisfies $\gamma_t \approx (1 + \alpha_t)\widetilde{\eta}_t$ and $\eta_t = \widetilde{\eta}_t$, generating a 3-sequence algorithm as in the original form of Algorithm 2.

Table 1: Example methods covered by the generic AGD template.

| Algorithm | Weights ($\alpha_t$) | Step-size ($\eta_t, \gamma_t$) | |
|---|---|---|---|
| **AdaGrad** | N/A | $\eta_t = \widetilde{\eta}_t$, | $\gamma_t = \eta_t$ |
| **AdaGrad w/ Averaging** | $\alpha_t = \frac{2}{t+1}$ or $\frac{1}{t}$ | $\eta_t = \alpha_t \widetilde{\eta}_t$, | $\gamma_t = 0$ |
| **Adaptive RSAG**(Ghadimi & Lan, 2016) | $\alpha_t = \frac{2}{t+1}$ | $\eta_t = \widetilde{\eta}_t$, | $\gamma_t = (1 + \alpha_t)\eta_t$ |
| **AcceleGrad**(Levy et al., 2018) | $\alpha_t = \frac{2}{t+1}$ | $\eta_t \approx \frac{1}{\alpha_t}\widetilde{\eta}_t$, | $\gamma_t \approx \widetilde{\eta}_t$ |

Before moving on to convergence results, we have an interesting observation concerning time-scale difference between step sizes for AdaGrad, AdaGrad with averaging and RSAG. Precisely, $\gamma_t$ is always *only* a constant factor away from $\eta_t$. This phenomenon has an immediate connection to acceleration in the convex realm, we will independently discuss at the end of this section.

Having defined instances of Algorithm 2, we will present high probability convergence rates for them. Similar to Eq. (10) for AdaGrad, we first define a departure point of similar structure in Proposition 5.1, then apply Proposition 4.1 and 4.2 in the same spirit as before to finalize the bound.

Let us define $\bar{g}_t = \nabla f(\bar{x}_t)$ and $g_t = \nabla f(\bar{x}_t; z_t) = \bar{g}_t + \xi_t$. Following the notation in (Ghadimi & Lan, 2016), let $\Gamma_t = (1 - \alpha_t)\Gamma_{t-1}$ with $\Gamma_1 = 1$. Now, we can begin with the departure point of the proof, which is due to Ghadimi & Lan (2016).

**Proposition 5.1.** *Let $x_t$ be generated by Algorithm 2 where $g_t = \nabla f(\bar{x}_t; z_t) = \bar{g}_t + \xi_t$ and $\bar{g}_t = \nabla f(\bar{x}_t)$. Then, it holds that*

$$\sum_{t=1}^{T} \|\bar{g}_t\|^2 \leq \frac{\Delta_{max} + 2L}{\eta_T} + \frac{L}{2\eta_T} \sum_{t=1}^{T} \underbrace{\left[\sum_{k=t}^{T}(1 - \alpha_k)\Gamma_k\right] \frac{\alpha_t}{\Gamma_t}}_{(*)} \frac{(\eta_t - \gamma_t)^2}{\alpha_t^2} \|g_t\|^2 + \underbrace{\sum_{t=1}^{T} -\langle \bar{g}_t, \xi_t \rangle}_{(**)}.$$

Next, we will deliver the complementary bound on term $(*)$ in Proposition 5.1.

**Proposition 5.2.** *Using the recursive definition of $\Gamma$, we have*

$$\left[\sum_{k=t}^{T}(1 - \alpha_k)\Gamma_k\right] \frac{\alpha_t}{\Gamma_t} \leq \begin{cases} 2 & \text{if } \alpha_t = \frac{2}{t+1}; \\ \log(T+1) & \text{if } \alpha_t = \frac{1}{t}. \end{cases}$$

Finally, we present the high probability convergence rates for AdaGrad with averaging and RSAG.

**Theorem 5.1.** *Let $x_t$ be the sequence generated by AdaGrad with averaging or adaptive RSAG. Under Assumptions 2, 6, 7, for $\Delta_{max} \leq O\left(\Delta_1 + L\log(T) + \sigma^2 \log(1/\delta)\right)$, with probability $1 - 8\log(T)\delta$,*

$$\frac{1}{T}\sum_{t=1}^{T} \|\bar{g}_t\|^2 \leq \frac{G_0(\Delta_{max} + 3L + L\log(\max\{1, G_0^{-2}\} + \tilde{G}^2 T)) + 3(G^2 + G\tilde{G})\log(1/\delta))}{T}$$

$$+ \frac{\tilde{G}(\Delta_{max} + 3L + L\log(\max\{1, G_0^{-2}\} + \tilde{G}^2 T)) + 2G\sigma\sqrt{\log(1/\delta)}}{\sqrt{T}}.$$

**A discussion on acceleration and nonconvex analysis:** AGD and its variants are able to converge at the fast rate of $\mathcal{O}(1/T^2)$ (Nesterov, 2003) for smooth, convex objectives, while matching the convergence rate of GD in the nonconvex landscapes. In fact, the mechanism that allows them to converge faster could even restrict their performance when convexity assumption is lifted. We will conclude this section with a brief discussion on this phenomenon.

As we mentioned previously, step-sizes for AdaGrad, its averaging variant and RSAG have the same time scale up to a constant factor. However, AcceleGrad, an accelerated algorithm, has a time-scale difference of $\mathcal{O}(t)$ between $\eta_t$ and $\gamma_t$, and it runs with a modified step-size of $\eta_t = (1 + \sum_{k=1}^{t} \alpha_k^{-2}\|g_t\|^2)^{-1/2}$. This scale difference is not possible to handle with standard approaches or our proposed analysis. If we look at second term in Proposition 5.1, it roughly evaluates to

$$\frac{L}{2\eta_T} \sum_{t=1}^{T} \frac{\eta_t^2}{\alpha_t^4}\|g_t\|^2,$$

where each summand is $\mathcal{O}(t^4)$ orders of magnitude larger compared to other methods. A factor of $\alpha_t^{-2} = t^2$ is absorbed by the modified step-size, but this term still grows faster than we can manage. We aim to understand it further and improve upon in our future attempts.

## 6 CONCLUSIONS

We propose a simple and modular high probability analysis for a class of AdaGrad-type algorithms. Bringing AdaGrad into the focus, we show that our new analysis techniques goes beyond and generalizes to the accelerated gradient template (Algorithm 2) which individually recovers AdaGrad, AdaGrad with averaging and adaptive version of RSAG (Ghadimi & Lan, 2016). By proposing a modification over standard analysis and relying on concentration bounds for martingales, we achieve high probability convergence bounds for the aforementioned algorithms *without* requiring the knowledge of smoothness $L$ and variance $\sigma$ while having best-known dependence of $\log(1/\delta)$ on $\delta$. To our knowledge, this is the first such result for adaptive methods, including AdaGrad.

## ACKNOWLEDGMENTS

This project has received funding from the European Research Council (ERC) under the European Union's Horizon 2020 research and innovation programme (grant agreement no 725594 - time-data) K.Y. Levy acknowledges support from the Israel Science Foundation (grant No. 447/20).

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

## A APPENDIX

### A.1 PROOF OF TECHNICAL LEMMAS IN SECTION 4.1

**Lemma 4.1.** *Let $a_1, ..., a_n$ be a sequence of non-negative real numbers. Then, it holds that*

$$\sqrt{\sum_{i=1}^{n} a_i} \leq \sum_{i=1}^{n} \frac{a_i}{\sqrt{\sum_{k=1}^{i} a_k}} \leq 2\sqrt{\sum_{i=1}^{n} a_i}$$

*Proof.* For the proof of first iniequality, please check Appendix A.4 of Levy et al. (2018), while that of the second one can be found at Appendix B of McMahan & Streeter (2010), which corresponds to their Lemma 5. □

**Lemma 4.2.** *Let $a_1, ..., a_n$ be a sequence of non-negative real numbers. Then, it holds that*

$$\sum_{i=1}^{n} \frac{a_i}{\sum_{k=1}^{i} a_i} \leq 1 + \log\left(1 + \sum_{i=1}^{n} a_i\right)$$

*Proof.* We will follow the proof steps of Levy et al. (2018) with a slight modification. The proof is due to induction.

For the base case of $n = 1$:

$$\frac{a_1}{a_1} = 1 \leq 1 + \log(1 + a_1)$$

Assume that the statement holds up to and including $n - 1 > 1$. Then, for $n$:

$$\sum_{i=1}^{n} \frac{a_i}{\sum_{j=1}^{i} a_j} \leq 1 + \log\left(1 + \sum_{i=1}^{n-1} a_i\right) + \frac{a_n}{\sum_{i=1}^{n} a_i} \overset{?}{\leq} 1 + \log\left(1 + \sum_{i=1}^{n} a_i\right)$$

We want to show that for any $a_n$, the second inequality with the question mark (?) holds. Let us define $x = \frac{a_n}{\sum_{i=1}^{n-1} a_i}$. Focusing on the second inequality and re arranging the terms we get,

$$\frac{a_n}{\sum_{i=1}^{n} a_i} \leq \log\left(\frac{1 + \sum_{i=1}^{n} a_i}{1 + \sum_{i=1}^{n-1} a_i}\right)$$

$$= \log\left(1 + \frac{a_n}{1 + \sum_{i=1}^{n-1} a_i}\right)$$

$$\leq \log\left(1 + \frac{a_n}{\sum_{i=1}^{n-1} a_i}\right)$$

Notice that

$$\frac{a_n}{\sum_{i=1}^{n} a_i} = \frac{a_n}{\sum_{i=1}^{n-1} a_i} \cdot \frac{\sum_{i=1}^{n-1} a_i}{\sum_{i=1}^{n} a_i} = \frac{a_n}{\sum_{i=1}^{n-1} a_i} \cdot \frac{1}{\frac{\sum_{i=1}^{n} a_i}{\sum_{i=1}^{n-1} a_i}} = \frac{a_n}{\sum_{i=1}^{n-1} a_i} \cdot \frac{1}{\left(1 + \frac{a_n}{\sum_{i=1}^{n-1} a_i}\right)}$$

$$= x\frac{1}{1 + x}$$

Combining both expressions,

$$\frac{x}{1 + x} \leq \log(1 + x)$$

which *always* holds whenever $x \geq 0$. □

**Lemma 4.3** (Lemma 3 in Kakade & Tewari (2008)). *Let $X_t$ be a martingale difference sequence such that $|X_t| \leq b$. Let us also define*

$$\mathbf{Var}_{t-1}(X_t) = \mathbf{Var}\left(X_t \mid X_1, ..., X_{t-1}\right) = \mathbb{E}\left[X_t^2 \mid X_1, ..., X_{t-1}\right],$$

*and define $V_T = \sum_{t=1}^{T} \mathbf{Var}_{t-1}(X_t)$ as the sum of variances. For $\delta < 1/e$ and $T \geq 3$, it holds that*

$$\mathbb{P}\left(\sum_{t=1}^{T} X_t > \max\left\{2\sqrt{V_T}, 3b\sqrt{\log(1/\delta)}\right\}\sqrt{\log(1/\delta)}\right) \leq 4\log(T)\delta \qquad (14)$$

*Proof.* The proof of this lemma could be found at the beginning of the Appendix section of Kakade & Tewari (2008), which is their Lemma 3 in the main text.

$\square$

## A.2 Proofs in Section 4.3

**Theorem 4.1.** *Let $x_t$ be generated by Algorithm 1 with $G_0 = 0$ for simplicity. Then, it holds that*

$$\frac{1}{T}\sum_{t=1}^{T}\|\nabla f(x_t)\|^2 \leq O\left(\frac{(\Delta_1 + L)^2}{T}\right).$$

*Proof (Theorem 4.1).* Setting $\bar{g}_t = \nabla f(x_t)$, deterministic counterpart of Eq. (9) becomes

$$\sum_{t=1}^{T}\|\bar{g}_t\|^2 \ \leq \ \frac{\Delta_{\max}}{\eta_T} + \frac{L}{2}\sum_{t=1}^{T}\eta_t\|\bar{g}_t\|^2 \ \leq \ (\Delta_{\max} + L)\sqrt{\sum_{t=1}^{T}\|\bar{g}_t\|^2},$$

where we obtain the final inequality using Lemma 4.1. Now, we show that $\Delta_{T+1}$ is bounded for any $T$. Using descent lemma and the update rule for $x_t$,

$$f(x_{t+1}) - f(x_t) \leq \langle \bar{g}_t, x_{t+1} - x_t \rangle + \frac{L}{2}\|x_{t+1} - x_t\|^2$$

$$\leq -\eta_t\|\bar{g}_t\|^2 + \frac{L\eta_t^2}{2}\|\bar{g}_t\|^2$$

Summing over $t \in [T]$, telescoping function values and re-arranging right-hand side,

$$f(x_{T+1}) - f(x_t) \leq \sum_{t=1}^{T}\left(\frac{L\eta_t}{2} - 1\right)\eta_t\|\bar{g}_t\|^2$$

$$f(x_{T+1}) - f(x^*) \leq f(x_1) - f(x^*) + \sum_{t=1}^{T}\left(\frac{L\eta_t}{2} - 1\right)\eta_t\|\bar{g}_t\|^2$$

where $x^* = \arg\min_{x \ in\mathbb{R}^d} f(x)$. Now, define $t_0 = \max\left\{t \in [T] \mid \eta_t > \frac{2}{L}\right\}$, such that $\left(\frac{L\eta_t}{2} - 1\right) \leq 0$ for any $t > t_0$. Then,

$$f(x_{T+1}) - f(x^*) \leq \Delta_1 + \sum_{t=1}^{t_0}\left(\frac{L\eta_t}{2} - 1\right)\eta_t\|\bar{g}_t\|^2 + \sum_{t=t_0+1}^{T}\left(\frac{L\eta_t}{2} - 1\right)\eta_t\|\bar{g}_t\|^2$$

$$\leq \Delta_1 + \frac{L}{2}\sum_{t=1}^{t_0}\eta_t^2\|\bar{g}_t\|^2 \qquad\qquad \text{(Lemma 4.2)}$$

$$\leq \Delta_1 + \frac{L}{2}\left(1 + \log\left(1 + \sum_{t=1}^{t_0}\|\bar{g}_t\|^2\right)\right) \qquad\qquad \text{(Definition of } \eta_t)$$

$$\leq \Delta_1 + \frac{L}{2}\left(1 + \log\left(1 + \frac{1}{\eta_{t_0}^2}\right)\right) \qquad\qquad \text{(Definition of } t_0)$$

$$\leq \Delta_1 + \frac{L}{2}\left(1 + \log\left(1 + \frac{L^2}{4}\right)\right),$$

where we use the definition of $t_0$ and Lemma 4.2 for the last inequality. Since this is true for any $T$, the bound holds for $\Delta_{\max}$ such that $\Delta_{\max} \leq \Delta_1 + \frac{L}{2}\left(1 + \log\left(L^2/4\right)\right)$. Now, define $X = \sqrt{\sum_{t=1}^{T}\|\bar{g}_t\|^2}$, then the original expression reduces to $X^2 \leq (\Delta_{\max} + L)X$. Solving for $X$ trivially yields

$$X \leq (\Delta_{\max} + L) \implies X^2 = \sum_{t=1}^{T}\|\bar{g}_t\|^2 \leq (\Delta_{\max} + L)^2.$$

Plugging in the bound for $\Delta_{\max}$ and dividing by $T$ gives,

$$\frac{1}{T}\sum_{t=1}^{T}\|\nabla f(x_t)\|^2 \leq \frac{\left(\Delta_1 + \frac{L}{2}\left(3 + \log\left(1 + \frac{L^2}{4}\right)\right)\right)^2}{T}$$

$\square$

**Theorem 4.2.** *Let $x_t$ be the sequence of iterates generated by AdaGrad. Under Assumptions 2, 6, 7, for $\Delta_{max} \leq O\left(\Delta_1 + L\log(T) + \sigma^2\log(1/\delta)\right)$, with probability at least $1 - 8\log(T)\delta$,*

$$\frac{1}{T}\sum_{t=1}^{T}\|\bar{g}_t\|^2 \leq \frac{(\Delta_{max} + L)G_0 + 3(G^2 + G\tilde{G})\log(1/\delta)}{T} + \frac{(\Delta_{max} + L)\tilde{G} + 2G\sigma\sqrt{\log(1/\delta)}}{\sqrt{T}}$$

*Proof (Theorem 4.2).* Define $\bar{g}_t = \nabla f(x_t)$ and $g_t = \nabla f(x_t; z_t) = \bar{g}_t + \xi_t$. By Eq. (10),

$$\sum_{t=1}^{T}\|\bar{g}_t\|^2 \leq \frac{\Delta_{\max}}{\eta_T} + \sum_{t=1}^{T} -\langle\bar{g}_t, \xi_t\rangle + \frac{L}{2}\sum_{t=1}^{T}\eta_t\|g_t\|^2$$

Invoking Lemma 4.1 and plugging the bound for the term $(**)$ from Proposition 4.1 we achieve with probability $1 - 4\log(T)\delta$,

$$\sum_{t=1}^{T}\|\bar{g}_t\|^2 \leq (\Delta_{\max} + L)\sqrt{G_0^2 + \sum_{t=1}^{T}\|g_t\|^2} + 2\sigma\sqrt{\log(1/\delta)}\sqrt{\sum_{t=1}^{T}\|\bar{g}_t\|^2} + 3(G^2 + G\tilde{G})\log(1/\delta)$$

$$\leq (\Delta_{\max} + L)\sqrt{G_0^2 + \tilde{G}^2 T} + 2G\sigma\sqrt{\log(1/\delta)}\sqrt{T} + 3(G^2 + G\tilde{G})\log(1/\delta)$$

$$\leq (\Delta_{\max} + L)G_0 + 3(G^2 + G\tilde{G})\log(1/\delta) + \left[(\Delta_{\max} + 2L)\tilde{G} + 2G\sigma\sqrt{\log(1/\delta)}\right]\sqrt{T}$$

Dividing both sides by T, we achieve the bound,

$$\frac{1}{T}\sum_{t=1}^{T}\|\bar{g}_t\|^2 \leq \frac{(\Delta_{\max} + L)G_0 + 3(G^2 + G\tilde{G})\log(1/\delta)}{T} + \frac{(\Delta_{\max} + L)\tilde{G} + 2G\sigma\sqrt{\log(1/\delta)}}{\sqrt{T}}$$

Now, we will incorporate the high probability bound for $\Delta_{\max}$ to complete the convergence proof. Essentially, we are interested in scenarios in which both the statement of Proposition 4.1 and the statement of Proposition 4.2 holds, simultaneously, with high probability. Formally, let the statement of Proposition 4.1 be denoted as event $A$ and the statement of Proposition 4.2 as event $B$. We have already proven that

$$\mathbb{P}(A) \geq 1 - 4\log(T)\delta \qquad \& \qquad \mathbb{P}(B) \geq 1 - 4\log(T)\delta$$

What we want to obtain is a lower bound to $\mathbb{P}(A \cap B)$, which is

$$\begin{aligned}\mathbb{P}(A \cap B) &= \mathbb{P}(A) + \mathbb{P}(B) - \mathbb{P}(A \cup B)\\ &\geq 1 - 4\log(T)\delta + 1 - 4\log(T)\delta - \mathbb{P}(A \cup B)\\ &\geq 2 - 8\log(T)\delta - 1 = 1 - 8\log(T)\delta,\end{aligned}$$

which is the best we could do due to the unknown extent of dependence between events $A$ and $B$. Hence, integrating the results of Proposition 4.2, with probability at least $1 - 8\log(T)\delta$,

$$\frac{1}{T}\sum_{t=1}^{T}\|\bar{g}_t\|^2 \leq \frac{(\Delta_{\max} + L)G_0 + 3(G^2 + G\tilde{G})\log(1/\delta)}{T} + \frac{(\Delta_{\max} + L)\tilde{G} + 2G\sigma\sqrt{\log(1/\delta)}}{\sqrt{T}}$$

where

$$\Delta_{\max} \leq \Delta_1 + 2L \left( 1 + \log \left( \max\{1, G_0^2\} + \tilde{G}^2 T \right) \right) + G_0^{-1}(3(G^2 + G\tilde{G}) + \sigma^2)\log(1/\delta) + G_0^{-1}(2G^2 + G\tilde{G})$$

$\square$

**Proposition 4.1.** *Using Lemma 4.3, with probability $1 - 4\log(T)\delta$ with $\delta < 1/e$, we have*

$$\sum_{t=1}^{T} -\langle \bar{g}_t, \xi_t \rangle \leq 2\sigma \sqrt{\log(1/\delta)} \sqrt{\sum_{t=1}^{T} \|\bar{g}_t\|^2} + 3(G^2 + G\tilde{G})\log(1/\delta).$$

*Proof.* We have to show that the random variable $-\langle \bar{g}_t, \xi_t \rangle$ is a martingale difference sequence and satisfies the conditions in Lemma 4.3. Let us define $\mathcal{F}_t = \sigma(\xi_t, ..., \xi_1)$ as the $\sigma$-algebra generated by randomness up to, and including $\xi_t$. Notice that $\mathcal{F}_t$ is the natural filtration of $-\langle \bar{g}_t, \xi_t \rangle$. Then, we need to show that

1. $-\langle \bar{g}_t, \xi_t \rangle$ is integrable,

2. *martingale (difference) property* holds, $\mathbb{E}\left[ -\langle \bar{g}_t, \xi_t \rangle | \mathcal{F}_{t-1} \right] = 0$.

First off, we show that $-\langle \bar{g}_t, \xi_t \rangle$ is integrable:

$$\begin{aligned}
\mathbb{E}\left[ |\langle \bar{g}_t, \xi_t \rangle| \right] &\leq \mathbb{E}\left[ \|\bar{g}_t\| \|\xi_t\| \right] \\
&= \mathbb{E}\left[ \|\bar{g}_t\|^2 + \|\xi_t\|^2 \right] \\
&\leq G^2 + \mathbb{E}\left[ \mathbb{E}\left[ \|\xi_t\| | \mathcal{F}_{t-1} \right] \right] \\
&\leq G^2 + \sigma^2 < +\infty,
\end{aligned}$$

where the second inequality is due to towering property of expectation. Then, the maritngale property:

$$\begin{aligned}
\mathbb{E}\left[ -\langle \bar{g}_t, \xi_t \rangle | \mathcal{F}_{t-1} \right] &= -\langle \bar{g}_t, \mathbb{E}\left[ \xi_t | \mathcal{F}_{t-1} \right] \rangle \\
&= -\langle \bar{g}_t, 0 \rangle = 0
\end{aligned}$$

Before applying Lemma 4.3, we need to verify that $|-\langle \bar{g}_t, \xi_t \rangle|$ is bounded:

$$|-\langle \bar{g}_t, \xi_t \rangle| = |-\langle \bar{g}_t, g_t - \bar{g}_t \rangle| = \left| \|\bar{g}_t\|^2 - \langle \bar{g}_t, g_t \rangle \right| \leq \|\bar{g}_t\|^2 + |-\langle \bar{g}_t, g_t \rangle| \leq \|\bar{g}_t\|^2 + \|\bar{g}_t\| \|g_t\| \leq G^2 + G\tilde{G},$$

where we used $G$-Lipschitzness of $f$ and almost sure boundedness of stochastic gradients $g_t$. Now, we are able make the high probability statement. By Lemma 4.3, with probability $1 - 4\log(T)\delta$ for $\delta < 1/e$, we have

$$\begin{aligned}
\sum_{t=1}^{T} -\langle \bar{g}_t, \xi_t \rangle &\leq \max \left\{ 2\sqrt{\sum_{t=1}^{T} \mathbb{E}\left[ \langle \bar{g}_t, \xi_t \rangle^2 | x_t \right]}, 3(G^2 + G\tilde{G})\sqrt{\log(1/\delta)} \right\} \sqrt{\log(1/\delta)} \\
&\overset{(1)}{\leq} \sqrt{\log(1/\delta)} \left( 2\sqrt{\sum_{t=1}^{T} \mathbb{E}\left[ \|\bar{g}_t\|^2 \|\xi_t\|^2 | x_t \right]} + 3(G^2 + G\tilde{G})\sqrt{\log(1/\delta)} \right) \\
&\overset{(2)}{\leq} \sqrt{\log(1/\delta)} \left( 2\sqrt{\sigma^2 \sum_{t=1}^{T} \|\bar{g}_t\|^2} + 3(G^2 + G\tilde{G})\sqrt{\log(1/\delta)} \right) \\
&\leq 2\sigma \sqrt{\log(1/\delta)} \sqrt{\sum_{t=1}^{T} \|\bar{g}_t\|^2} + 3(G^2 + G\tilde{G})\log(1/\delta)
\end{aligned}$$

where we used Cauchy-Schwarz inequality for the inner product to obtain inequality (1) and bounded variance assumption to obtain (2). $\square$

**Proposition 4.2.** *Let $x_t$ be generated by AdaGrad for $G_0 > 0$. With probability at least $1 - 4\log(t)\delta$,*

$$\Delta_{t+1} \leq \Delta_1 + 2L\left(1 + \log\left(G_0^2 + \tilde{G}^2 t\right)\right) + G_0^{-1}(M_1 + \sigma^2)\log(1/\delta) + M_2,$$

*where $M_1 = 3(G^2 + G\tilde{G})$ and $M_2 = G_0^{-1}(2G^2 + G\tilde{G})$.*

*Proof.* We will handle this bound in two cases. First, we show the bound for AdaGrad, and then for the remaining two algorithms. Indeed, the bounds for the two cases differ by a factor of constants., hence we will use the larger bound for all three algorithms.

**Case 1  (AdaGrad)**
Let $\bar{g}_t = \nabla f(x_t)$ and $g_t = \nabla f(x_t; z_t)$, such that $\bar{g}_t = g_t + \xi_t$. Then, by smoothness

$$f(x_{t+1}) - f(x_t) \leq -\eta_t \langle \bar{g}_t, g_t \rangle + \frac{L\eta_t^2}{2}\|g_t\|^2$$

$$= -\eta_t\|\bar{g}_t\|^2 - \eta_t\langle \bar{g}_t, \xi_t \rangle + \frac{L\eta_t^2}{2}\|g_t\|^2$$

Defining $x^* = \min_{x \in \mathbb{R}^d} f(x)$ as the global minimizer of $f$ and summing over $t \in [T]$,

$$f(x_{T+1}) - f(x^*) \leq f(x_1) - f(x^*) + \underbrace{\sum_{t=1}^{T} -\eta_t\|\bar{g}_t\|^2}_{(A)} + \underbrace{\frac{L}{2}\sum_{t=1}^{T}\eta_t^2\|g_t\|^2}_{(B)} + \underbrace{\sum_{t=1}^{T} -\eta_t\langle \bar{g}_t, \xi_t \rangle}_{(C)} \quad (15)$$

**Term (A)**   At this point, we will keep this term as it will be coupled with the sum-of-conditional-variances term which will be obtained through martingale concentration.

**Term (B)**

$$\frac{L}{2}\sum_{t=1}^{T}\eta_t^2\|g_t\|^2 = \frac{L}{2}\sum_{t=1}^{T}\frac{\|g_t\|^2}{G_0^2 + \sum_{i=1}^{t}\|g_i\|^2} \qquad \text{(Lemma 4.2)}$$

$$\leq \frac{L}{2}\left(1 + \log\left(\max\{1, G_0^2\} + \sum_{t=1}^{T}\|g_t\|^2\right)\right) \qquad \text{(Bounded gradients)}$$

$$\leq \frac{L}{2}\left(1 + \log\left(\max\{1, G_0^2\} + \tilde{G}^2 T\right)\right)$$

**Bounding term (C)**

$$\sum_{t=1}^{T} -\eta_t\langle \bar{g}_t, \xi_t \rangle \leq \underbrace{\sum_{t=1}^{T} -\eta_{t-1}\langle \bar{g}_t, \xi_t \rangle}_{(C.1)} + \underbrace{\sum_{t=1}^{T}(\eta_{t-1} - \eta_t)\langle \bar{g}_t, \xi_t \rangle}_{(C.2)}$$

We will make use of Lemma 4.3 to achieve high probability bounds on term (C.1). To do so, we need to prove that $X_t = -\eta_{t-1}\langle \bar{g}_t, \xi_t \rangle$ is a martingale difference sequence and validate some of its properties:

1. $-\eta_{t-1}\langle \bar{g}_t, \xi_t \rangle$ is absolutely integrable:

$$\mathbb{E}[|-\eta_{t-1}\langle \bar{g}_t, \xi_t \rangle|] \leq G_0^{-1}\mathbb{E}[|\langle \bar{g}_t, \xi_t \rangle|]$$

$$\leq G_0^{-1}\mathbb{E}[\|\bar{g}_t\|^2 + \|\xi_t\|^2]$$

$$\leq G_0^{-1}(G^2 + \sigma^2) < +\infty$$

2. $-\eta_{t-1}\langle \bar{g}_t, \xi_t \rangle$ is adapted to its natural filtration $\mathcal{F}_t = \sigma(\xi_1, ..., \xi_t)$

3. It satisfies the martingale (difference) property:
$$\mathbb{E}\left[-\eta_{t-1}\langle\bar{g}_t,\xi_t\rangle \mid \mathcal{F}_{t-1}\right] = -\eta_{t-1}\langle\bar{g}_t,\mathbb{E}\left[\xi_t \mid \mathcal{F}_{t-1}\right]\rangle = 0$$

4. $X_t = -\eta_{t-1}\langle\bar{g}_t,\xi_t\rangle$ is bounded:
$$-\eta_{t-1}\langle\bar{g}_t,\xi_t\rangle \le G_0^{-1}\,|\langle\bar{g}_t,\xi_t\rangle| \le G_0^{-1}(\|\bar{g}_t\|^2 + \|\bar{g}_t\|\|g_t\|) \le G_0^{-1}(G^2 + G\tilde{G})$$

5. Conditional variance of $X_t = -\eta_{t-1}\langle\bar{g}_t,\xi_t\rangle$:
$$\begin{aligned}
\mathbf{Var}_{t-1}(X_t) &= \mathbb{E}\left[\left(\eta_{t-1}\langle\bar{g}_t,\xi_t\rangle\right)^2 \mid \mathcal{F}_{t-1}\right] \\
&\le G_0^{-2}\mathbb{E}\left[\left(\langle\bar{g}_t,\xi_t\rangle\right)^2 \mid \mathcal{F}_{t-1}\right] \\
&\le G_0^{-2}\|\bar{g}_t\|^2\mathbb{E}\left[\|\xi_t\|^2 \mid \mathcal{F}_{t-1}\right] \\
&\le G_0^{-2}\sigma^2\|\bar{g}_t\|^2
\end{aligned}$$

**Term (C.1)**   Now, we are at a position to apply Lemma 4.3 on term (C.1). With probability $1 - 4\log(T)\delta$,

$$\begin{aligned}
\sum_{t=1}^{T}-\eta_{t-1}\langle\bar{g}_t,\xi_t\rangle &\le \max\left\{2\sqrt{\sum_{t=1}^{T}\mathbb{E}\left[\left(\eta_{t-1}\langle\bar{g}_t,\xi_t\rangle\right)^2 \mid \mathcal{F}_{t-1}\right]}, 3G_0^{-1}(G^2+G\tilde{G})\sqrt{\log(1/\delta)}\right\}\sqrt{\log(1/\delta)} \\
&\le \max\left\{2\sqrt{\sum_{t=1}^{T}\sigma^2\eta_{t-1}^2\|\bar{g}_t\|^2}, 3G_0^{-1}(G^2+G\tilde{G})\sqrt{\log(1/\delta)}\right\}\sqrt{\log(1/\delta)} \\
&\le \underbrace{2\sigma\sqrt{\log(1/\delta)}\sqrt{\sum_{t=1}^{T}\eta_{t-1}^2\|\bar{g}_t\|^2}}_{(D)} + 3G_0^{-1}(G^2+G\tilde{G})\log(1/\delta)
\end{aligned}$$

**Term (C.2):**

$$\begin{aligned}
\sum_{t=1}^{T}(\eta_{t-1}-\eta_t)\langle\bar{g}_t,\xi_t\rangle &\le \sum_{t=1}^{T}(\eta_{t-1}-\eta_t)\,|\langle\bar{g}_t,\xi_t\rangle| \\
&\le (G^2+G\tilde{G})\sum_{t=1}^{T}(\eta_{t-1}-\eta_t) \\
&\le (G^2+G\tilde{G})\eta_0
\end{aligned}$$

**Terms (A) + (D):**   All the underbraced term but expression (D) either grows as $O\left(\log(T)\right)$, or is upper bounded by a constant. The worst-case growth of term (D) is $O(\sqrt{T})$, which we will keep under control via term (A).

$$\begin{aligned}
\text{(A) + (D)} &\le 2\sigma\sqrt{\log(1/\delta)}\sqrt{\sum_{t=1}^{T}\eta_{t-1}^2\|\bar{g}_t\|^2 - \sum_{t=1}^{T}\eta_t\|\bar{g}_t\|^2} \\
&\le 2\sigma\sqrt{\log(1/\delta)}\sqrt{\sum_{t=1}^{T}\eta_{t-1}^2\|\bar{g}_t\|^2 - G_0\sum_{t=1}^{T}\eta_t^2\|\bar{g}_t\|^2} \\
&\le 2\sigma\sqrt{\log(1/\delta)}\sqrt{\sum_{t=1}^{T}\eta_{t-1}^2\|\bar{g}_t\|^2 - G_0\sum_{t=1}^{T}\eta_{t-1}^2\|\bar{g}_t\|^2 + G_0\sum_{t=1}^{T}(\eta_{t-1}^2-\eta_t^2)\|\bar{g}_t\|^2} \\
&\le 2\sigma\sqrt{\log(1/\delta)}\sqrt{\sum_{t=1}^{T}\eta_{t-1}^2\|\bar{g}_t\|^2 - G_0\sum_{t=1}^{T}\eta_{t-1}^2\|\bar{g}_t\|^2 + G_0 G^2\eta_0^2}
\end{aligned}$$

In order to characterize the growth of this expression, let us define $f(x) = 2\sigma\sqrt{\log(1/\delta)}\sqrt{x} - G_0 x$, which is a concave function as its second derivative is non-positive. Now, looking at derivative of $f$,

$$\frac{d}{dx}f(x) = \frac{\sigma\sqrt{\log(1/\delta)}}{\sqrt{x}} - G_0,$$

which is 0 at $x = G_0^{-2}\sigma^2\log(1/\delta)$. This is indeed the point at which the function attains its maximum. For the final step of the proof, we define $Z_T = \sum_{t=1}^{T}\eta_{t-1}^2\|\bar{g}_t\|^2$. Then,

$$\begin{aligned}
\text{(A)} + \text{(D)} &\leq f(Z_T) + G_0 G^2 \eta_0^2 \\
&\leq f(G_0^{-2}\sigma^2\log(1/\delta)) + G_0 G^2 \eta_0^2 \\
&= G_0^{-1}\sigma^2\log(1/\delta) + G_0 G^2 \eta_0^2
\end{aligned}$$

**Final bound**   Plugging all the expression together and setting $\eta_0 = \eta_1$, with probability at least $1 - 4\log(T)\delta$,

$$\begin{aligned}
f(x_{T+1}) - f(x^*) \leq{}& f(x_1) - f(x^*) + \frac{L}{2}\left(1 + \log\left(\max\{1, G_0^2\} + \tilde{G}^2 T\right)\right) \\
&+ G_0^{-1}(3(G^2 + G\tilde{G}) + \sigma^2)\log(1/\delta) \\
&+ G_0^{-1}(2G^2 + G\tilde{G})
\end{aligned}$$

Since this result holds for any T, to make it consistent with the statement of the proposition, we re-state the bound with $t$,

$$\begin{aligned}
f(x_{t+1}) - f(x^*) \leq{}& f(x_1) - f(x^*) + \frac{L}{2}\left(1 + \log\left(\max\{1, G_0^2\} + \tilde{G}^2 t\right)\right) \\
&+ G_0^{-1}(3(G^2 + G\tilde{G}) + \sigma^2)\log(1/\delta) \\
&+ G_0^{-1}(2G^2 + G\tilde{G})
\end{aligned}$$

**Case 2   (AdaGrad w/ Averaging & RSAG)** Let us define stochastic gradient at $\bar{x}_t$ as $g_t = \nabla f(\bar{x}_t; z_t) = \bar{g}_t + \xi_t$ where $\bar{g}_t = \nabla f(\bar{x}_t)$. Then, by smoothness,

$$\begin{aligned}
f(x_{t+1}) - f(x_t) \leq{}& \langle\nabla f(x_t), x_{t+1} - x_t\rangle + \frac{L}{2}\|x_{t+1} - x_t\|^2 \\
={}& -\eta_t\langle\bar{g}_t, g_t\rangle - \eta_t\langle\nabla f(x_t) - \bar{g}_t, g_t\rangle + \frac{L\eta_t^2}{2}\|g_t\|^2 && \text{(Cauchy-Schwarz)} \\
={}& -\eta_t\|\nabla f(x_t) - \eta_t\langle\bar{g}_t, \xi_t\rangle + \nabla f(\bar{x}_t)\|\|g_t\| + \frac{L\eta_t^2}{2}\|g_t\|^2 && \text{(Smoothness)} \\
={}& -\eta_t\|\bar{g}_t\|^2 - \eta_t\langle\bar{g}_t, \xi_t\rangle + L\eta_t\|\bar{x}_t - x_t\|\|g_t\| + \frac{L\eta_t^2}{2}\|g_t\|^2 && \text{(Young's ineq.)} \\
={}& -\eta_t\|\bar{g}_t\|^2 - \eta_t\langle\bar{g}_t, \xi_t\rangle + \frac{L}{2}\|\bar{x}_t - x_t\|^2 + L\eta_t^2\|g_t\|^2
\end{aligned}$$

Using recursive expansion of $\|\bar{x}_t - x_t\|^2$ and summing over $t \in [T]$,

$$\begin{aligned}
\Delta_{T+1} \leq{}& \Delta_1 + \frac{L}{2}\sum_{t=1}^{T}\left[(1 - \alpha_t)\Gamma_t\sum_{k=1}^{t}\frac{\alpha_k}{\Gamma_k}\frac{(\eta_k - \gamma_k)^2}{\alpha_k^2}\|g_t\|^2\right] + \sum_{t=1}^{T}L\eta_t^2\|g_t\|^2 - \eta_t\|\bar{g}_t\|^2 - \eta_t\langle\bar{g}_t, \xi_t\rangle \\
\leq{}& \Delta_1 + \frac{L}{2}\sum_{t=1}^{T}\left[\sum_{k=t}^{T}(1 - \alpha_k)\Gamma_k\right]\frac{\alpha_t}{\Gamma_t}\frac{(\eta_k - \gamma_k)^2}{\alpha_k^2}\|g_t\|^2 + \sum_{t=1}^{T}L\eta_t^2\|g_t\|^2 - \eta_t\|\bar{g}_t\|^2 - \eta_t\langle\bar{g}_t, \xi_t\rangle
\end{aligned}$$

First, we plug in $\alpha_t = 2/(t+1)$ and invoke Proposition 5.2. Recognizing that $|\gamma_t - \eta_t| = \alpha_t \eta_t$ for both RSAG and AdaGrad with averaging,

$$\Delta_{T+1} \leq \Delta_1 + \underbrace{\sum_{t=1}^{T} -\eta_t \|\bar{g}_t\|^2}_{(A)} + \underbrace{2L \sum_{t=1}^{T} \eta_t^2 \|g_t\|^2}_{(B)} + \underbrace{\sum_{t=1}^{T} -\eta_t \langle \bar{g}_t, \xi_t \rangle}_{(C)}$$

Observe that this expression is the same as Eq. (15) up to replacing $\frac{L}{2}$ in term (B) of AdaGrad with $2L$. Hence, the same bounds hold up to incorporating the aforementioned change. With probability $1 - 4\log(T)\delta$,

$$f(x_{T+1}) - f(x^*) \leq f(x_1) - f(x^*) + 2L \left( 1 + \log \left( \max\{1, G_0^2\} + \tilde{G}^2 T \right) \right)$$
$$+ G_0^{-1}(3(G^2 + G\tilde{G}) + \sigma^2) \log(1/\delta)$$
$$+ G_0^{-1}(2G^2 + G\tilde{G})$$

Similarly, since this holds for any $T$, we re-state the results with $t$ for consistency,

$$f(x_{t+1}) - f(x^*) \leq f(x_1) - f(x^*) + 2L \left( 1 + \log \left( \max\{1, G_0^2\} + \tilde{G}^2 t \right) \right)$$
$$+ G_0^{-1}(3(G^2 + G\tilde{G}) + \sigma^2) \log(1/\delta)$$
$$+ G_0^{-1}(2G^2 + G\tilde{G})$$

$\square$

### A.3 PROOFS IN SECTION 5

**Proposition 5.1.** *Let $x_t$ be generated by Algorithm 2 where $g_t = \nabla f(\bar{x}_t; z_t) = \bar{g}_t + \xi_t$ and $\bar{g}_t = \nabla f(\bar{x}_t)$. Then, it holds that*

$$\sum_{t=1}^{T} \|\bar{g}_t\|^2 \leq \frac{\Delta_{max} + 2L}{\eta_T} + \frac{L}{2\eta_T} \sum_{t=1}^{T} \underbrace{\left[ \sum_{k=t}^{T} (1-\alpha_k)\Gamma_k \right] \frac{\alpha_t}{\Gamma_t}}_{(*)} \frac{(\eta_t - \gamma_t)^2}{\alpha_t^2} \|g_t\|^2 + \underbrace{\sum_{t=1}^{T} -\langle \bar{g}_t, \xi_t \rangle}_{(**)}.$$

*Proof.* This result is due to Ghadimi & Lan (2016); Lan (2020) up to introducing adaptive step-sizes. We follow their derivations in the deterministic setting and incorporate it with our high probability analysis. Then,

$$f(x_{t+1}) - f(x_t) \leq \langle \nabla f(x_t), x_{t+1} - x_t \rangle + \frac{L}{2} \|x_{t+1} - x_t\|^2$$
$$\leq -\eta_t \langle \nabla f(x_t), \nabla f(\bar{x}_t) + \xi_t \rangle + \frac{L\eta_t^2}{2} \|g_t\|^2$$
$$= -\eta_t \|\bar{g}_t\|^2 - \eta_t \langle \nabla f(\bar{x}_t), \xi_t \rangle - \eta_t \langle \nabla f(x_t) - \nabla f(\bar{x}_t), g_t \rangle + \frac{L\eta_t^2}{2} \|g_t\|^2$$
$$\leq -\eta_t \|\bar{g}_t\|^2 - \eta_t \langle \bar{g}_t, \xi_t \rangle + \frac{L}{2} \|\bar{x}_t - x_t\|^2 + L\eta_t^2 \|g_t\|^2$$

where we used descent lemma (Eq. (4)) in the first inequality, and update rule for $x_{t+1}$ in Algorithm 2, line 4 in the second inequality. For the last line, we use Cauchy-Schwarz, apply smoothnness definition in Eq. (3) and finally use Young's inequality. Let us define $\Delta_t = f(x_t) - \min_{x \in \mathbb{R}^d} f(x)$ and $\Delta_{\max} = \max_{t \in [T]} \Delta_t$. Dividing both sides by $\eta_t$, rearranging, and summing over $t = 1, ..., T$ we obtain,

$$\sum_{t=1}^{T} \|\bar{g}_t\|^2 \leq \sum_{t=1}^{T} \frac{1}{\eta_t} (\Delta_t - \Delta_{t+1}) + \frac{L}{2} \sum_{t=1}^{T} \frac{1}{\eta_t} \|\bar{x}_t - x_t\|^2 + L \sum_{t=1}^{T} \eta_t \|g_t\|^2 + \sum_{t=1}^{T} -\langle \bar{g}_t, \xi_t \rangle$$

Now, we express the term $\bar{x}_t - x_t$ recursively, as a function of gradient norms.

$$
\begin{aligned}
\bar{x}_t - x_t &= (1 - \alpha_t) \left[\tilde{x}_t - x_t\right] \\
&= (1 - \alpha_t) \left[\bar{x}_{t-1} - x_{t-1} + (\eta_{t-1} - \gamma_{t-1})g_{t-1}\right] \\
&= (1 - \alpha_t) \left[(1 - \alpha_{t-1})(\tilde{x}_{t-1} - x_{t-1}) + (\eta_{t-1} - \gamma_{t-1})g_{t-1}\right] \\
&= (1 - \alpha_t) \sum_{k=1}^{t-1} \left(\prod_{j=k+1}^{t-1} (1 - \alpha_j)\right) (\eta_k - \gamma_k)g_k \\
&= (1 - \alpha_t) \sum_{k=1}^{t-1} \frac{\Gamma_{t-1}}{\Gamma_k} (\eta_k - \gamma_k)g_k \\
&= (1 - \alpha_t)\Gamma_{t-1} \sum_{k=1}^{t-1} \frac{\alpha_k}{\Gamma_k} \frac{(\eta_k - \gamma_k)}{\alpha_k} g_k,
\end{aligned}
$$

Hence, by convexity of squared norm and (absolute) homogeneity of vector norms,

$$
\begin{aligned}
\|\bar{x}_t - x_t\|^2 &= \left\|(1 - \alpha_t)\Gamma_{t-1} \sum_{k=1}^{t-1} \frac{\alpha_k}{\Gamma_k} \frac{(\eta_k - \gamma_k)}{\alpha_k} g_k\right\|^2 \\
&\leq (1 - \alpha_t)^2 \Gamma_{t-1} \sum_{k=1}^{t-1} \frac{\alpha_k}{\Gamma_k} \frac{(\eta_k - \gamma_k)^2}{\alpha_k^2} \|g_k\|^2 \\
&\leq (1 - \alpha_t)\Gamma_t \sum_{k=1}^{t} \frac{\alpha_k}{\Gamma_k} \frac{(\eta_k - \gamma_k)^2}{\alpha_k^2} \|g_k\|^2
\end{aligned}
$$

Finally, we plug this in the original expression,

$$
\begin{aligned}
\sum_{t=1}^{T} \|\bar{g}_t\|^2 &\leq \sum_{t=1}^{T} \frac{1}{\eta_t} (\Delta_t - \Delta_{t+1}) + \frac{L}{2} \sum_{t=1}^{T} \left[(1 - \alpha_t)\frac{\Gamma_t}{\eta_t} \sum_{k=1}^{t} \frac{\alpha_k}{\Gamma_k} \frac{(\eta_k - \gamma_k)^2}{\alpha_k^2} \|g_k\|^2\right] \\
&\quad + L \sum_{t=1}^{T} \eta_t \|g_t\|^2 + \sum_{t=1}^{T} -\langle \bar{g}_t, \xi_t\rangle \\
&\leq \frac{\Delta_1}{\eta_1} + \sum_{t=1}^{T-1} (\frac{1}{\eta_{t+1}} - \frac{1}{\eta_t})\Delta_{t+1} + \frac{L}{2} \sum_{t=1}^{T} \left[\sum_{k=t}^{T}(1 - \alpha_k)\frac{\Gamma_k}{\eta_k}\right] \frac{\alpha_t}{\Gamma_t} \frac{(\eta_t - \gamma_t)^2}{\alpha_t^2} \|g_t\|^2 \\
&\quad + L \sum_{t=1}^{T} \frac{\|g_t\|^2}{\sqrt{G_0^2 + \sum_{k=1}^{t} \|g_k\|^2}} + \sum_{t=1}^{T} -\langle \bar{g}_t, \xi_t\rangle \\
&\leq \frac{\Delta_{\max}}{\eta_1} + \Delta_{\max} \sum_{t=1}^{T-1} (\frac{1}{\eta_{t+1}} - \frac{1}{\eta_t}) + \frac{L}{2} \sum_{t=1}^{T} \left[\sum_{k=t}^{T}(1 - \alpha_k)\frac{\Gamma_k}{\eta_k}\right] \frac{\alpha_t}{\Gamma_t} \frac{(\eta_t - \gamma_t)^2}{\alpha_t^2} \|g_t\|^2 \\
&\quad + 2L\sqrt{G_0^2 + \sum_{t=1}^{T} \|g_t\|^2} + \sum_{t=1}^{T} -\langle \bar{g}_t, \xi_t\rangle \\
&\leq \frac{\Delta_{\max} + 2L}{\eta_T} + \frac{L}{2\eta_T} \sum_{t=1}^{T} \underbrace{\left[\sum_{k=t}^{T}(1 - \alpha_k)\Gamma_k\right] \frac{\alpha_t}{\Gamma_t} \frac{(\eta_t - \gamma_t)^2}{\alpha_t^2} \|g_t\|^2}_{(*)} + \underbrace{\sum_{t=1}^{T} -\langle \bar{g}_t, \xi_t\rangle}_{(**)}.
\end{aligned}
$$

We rearranged the summations to obtain the second inequality, used the assumption that $\Delta_t \leq \Delta_{\max}$ for any $t$ together with Lemma 4.1, and we telescope the first summation on the right hand side to obtain the result.

$\square$

Next, we provide the proof for term $(*)$ in Proposition 5.1.

**Proposition 5.2.** *We have*

$$\left[\sum_{k=t}^{T}(1-\alpha_k)\Gamma_k\right]\frac{\alpha_t}{\Gamma_t} \leq \begin{cases} 2 & \text{if } \alpha_t = \frac{2}{t+1}; \\ \log(T+1) & \text{if } \alpha_t = \frac{1}{t}. \end{cases}$$

*Proof.* First, we begin with the weighted averaging setting, i.e., $\alpha_t = 2/(t+1)$. Using the recursive definition of $\Gamma$, one could easily show that for any $\alpha_t \in (0,1)$,

$$\sum_{k=1}^{t}\frac{\alpha_k}{\Gamma_k} = \frac{1}{\Gamma_t} \quad \Longrightarrow \quad \Gamma_t\sum_{k=1}^{t}\frac{\alpha_k}{\Gamma_k} = 1.$$

Defining $A_t = \sum_{i=1}^{t} i = \frac{t(t+1)}{2}$ and $A_0 = 1$, we have that $\alpha_t = \frac{t}{A_t}$ and

$$\Gamma_t = \prod_{i=1}^{t}(1-\alpha_i) = \prod_{i=1}^{t}(1-\frac{i}{A_i}) = \prod_{i=1}^{t}\frac{A_{i-1}}{A_i} = \frac{1}{A_t}$$

Hence, we can express term $(*)$ as

$$\left[\sum_{k=t}^{T}(1-\alpha_k)\Gamma_k\right]\frac{\alpha_t}{\Gamma_t} \leq \left[\sum_{k=t}^{T}\frac{1}{A_k}\right]t$$

$$= \left[2\sum_{k=t}^{T}\frac{1}{k(k+1)}\right]t$$

$$= \left[2\sum_{k=t}^{T}\frac{1}{k}-\frac{1}{k+1}\right]t$$

$$= 2\left(\frac{1}{t}-\frac{1}{T+1}\right)t$$

$$\leq 2$$

For the uniform averaging setting with $\alpha_t = \frac{1}{t}$, for $t > 1$,

$$\Gamma_t = \prod_{i=1}^{t}(1-\alpha_i) = \prod_{i=2}^{t}\frac{k-1}{k} = \frac{1}{k}$$

Hence, again for $t > 1$,

$$\left[\sum_{k=t}^{T}(1-\alpha_k)\Gamma_k\right]\frac{\alpha_t}{\Gamma_t} = \sum_{k=t}^{T}\frac{1}{k} \leq \sum_{k=2}^{T}\frac{1}{k} \leq \log(T+1),$$

where the last inequality is due to that fact that integral of $f(x) = 1/x$ over the range $[1,k]$ upper bounds the summation above.

$\square$

Finally, we conclude with the high probability convergence theorem for AdaGrad with averaging and RSAG,

**Theorem 5.1.** *Let $x_t$ be the sequence of iterates generated by AdaGrad with averaging or adaptive RSAG. Under Assumptions 2, 6, 7, for $\Delta_{max} \leq O\left(\Delta_1 + L\log(T) + \sigma^2\log(1/\delta)\right)$, with probability*

$1 - 8\log(T)\delta$,

$$\frac{1}{T}\sum_{t=1}^{T}\|\bar{g}_t\|^2 \le \frac{G_0\left(\Delta_{max} + 3L + L\log\left(\max\{1, G_0^{-2}\} + \tilde{G}^2 T\right)\right) + 3(G^2 + G\tilde{G})\log(1/\delta))}{T}$$
$$+ \frac{\tilde{G}\left(\Delta_{max} + 3L + L\log\left(\max\{1, G_0^{-2}\} + \tilde{G}^2 T\right)\right) + 2G\sigma\sqrt{\log(1/\delta)}}{\sqrt{T}}$$

*Proof.* Again by Proposition 5.1,

$$\sum_{t=1}^{T}\|\bar{g}_t\|^2 \le \frac{\Delta_{\max} + 2L}{\eta_T} + \frac{L}{2\eta_T}\sum_{t=1}^{T}\underbrace{\left[\sum_{k=t}^{T}(1-\alpha_k)\Gamma_k\right]\frac{\alpha_t}{\Gamma_t}\frac{(\eta_t - \gamma_t)^2}{\alpha_t^2}\|g_t\|^2}_{(*)} + \underbrace{\sum_{t=1}^{T}-\langle\bar{g}_t, \xi_t\rangle}_{(**)}.$$

Both for AdaGrad with averaging and adaptive RSAG, we use weighted averaging. Moreover, due to the particular step-size choices, $\eta_t - \gamma_t = \alpha_t\eta_t$ for both methods. Combining this observation with the previous expression we get

$$\sum_{t=1}^{T}\|\bar{g}_t\|^2 \le \frac{\Delta_{\max} + 2L}{\eta_T} + \frac{L}{2\eta_T}\sum_{t=1}^{T}\underbrace{\left[\sum_{k=t}^{T}(1-\alpha_k)\Gamma_k\right]\frac{\alpha_t}{\Gamma_t}\eta_t^2\|g_t\|^2}_{(*)} + \underbrace{\sum_{t=1}^{T}-\langle\bar{g}_t, \xi_t\rangle}_{(**)}.$$

We introduce the bounds in Proposition 5.2 and Proposition 4.1 for the respective marked term,

$$\sum_{t=1}^{T}\|\bar{g}_t\|^2 \le \frac{\Delta_{\max} + 2L + L\sum_{t=1}^{T}\eta_t^2\|g_t\|^2}{\eta_T} + 2\sigma\sqrt{\log(1/\delta)}\sqrt{\sum_{t=1}^{T}\|\bar{g}_t\|^2} + 3(G^2 + G\tilde{G})\log(1/\delta)$$

$$\le \frac{\Delta_{\max} + 3L + L\log\left(\max\{1, G_0^{-2}\} + \sum_{t=1}^{T}\|g_t\|^2\right)}{\eta_T}$$
$$+ 2G\sigma\sqrt{\log(1/\delta)}\sqrt{T} + 3(G^2 + G\tilde{G})\log(1/\delta)$$

$$\le \left(\Delta_{\max} + 3L + L\log\left(\max\{1, G_0^{-2}\} + \sum_{t=1}^{T}\|g_t\|^2\right)\right)\sqrt{G_0^2 + \sum_{t=1}^{T}\|g_t\|^2}$$
$$+ 2G\sigma\sqrt{\log(1/\delta)}\sqrt{T} + 3(G^2 + G\tilde{G})\log(1/\delta)$$

$$\le \left(\tilde{G}\left(\Delta_{\max} + 3L + L\log\left(\max\{1, G_0^{-2}\} + \tilde{G}^2 T\right)\right) + 2G\sigma\sqrt{\log(1/\delta)}\right)\sqrt{T}$$
$$+ G_0\left(\Delta_{\max} + 3L + L\log\left(\max\{1, G_0^{-2}\} + \tilde{G}^2 T\right)\right) + 3(G^2 + G\tilde{G})\log(1/\delta)$$

where we used Lemma 4.2 in the second inequality, while boundedness of $\bar{g}_t$ and almost sure boundedness of $g_t$ in the last line. Dividing both sides by T, and using the same argument as in the proof of Theorem 4.2, with probability at least $1 - 8\log(T)\delta$,

$$\frac{1}{T}\sum_{t=1}^{T}\|\bar{g}_t\|^2 \le \frac{G_0\left(\Delta_{\max} + 3L + L\log\left(\max\{1, G_0^{-2}\} + \tilde{G}^2 T\right)\right) + 3(G^2 + G\tilde{G})\log(1/\delta))}{T}$$
$$+ \frac{\tilde{G}\left(\Delta_{\max} + 3L + L\log\left(\max\{1, G_0^{-2}\} + \tilde{G}^2 T\right)\right) + 2G\sigma\sqrt{\log(1/\delta)}}{\sqrt{T}}$$

where
$$\Delta_{\max} \leq \Delta_1 + 2L \left(1 + \log\left(\max\{1, G_0^2\} + \tilde{G}^2 T\right)\right) + G_0^{-1}(3(G^2 + G\tilde{G}) + \sigma^2)\log(1/\delta) + G_0^{-1}(2G^2 + G\tilde{G})$$

$\square$

## B  NOISE ADAPTATION UNDER SUB-GAUSSIAN NOISE

In this section of the appendix, we present the proof of Theorems 4.3 and 4.4 along with the Lemmas that we will require in the proofs. We first prove a bound on $\Delta_{\max}$ and then show noise-adaptive rates for AdaGrad.

### B.1  HIGH PROBABILITY BOUNDS ON $\Delta_{\text{MAX}}$

We will argue about high probability bounds on objective sub-optimality under sub-Gaussian assumption. First, we will present Lemma 1 from Li & Orabona (2020), as well as our modified version of it that we use in our derivations.

**Lemma B.1** (Lemma 1 from Li & Orabona (2020)). *Let $Z_1, \cdots, Z_T$ be a martingale difference sequence (MDS) with respect to random vectors $\xi_1, \cdots, \xi_T$ and $Y_t$ be a sequence of random variables which is $\sigma(\xi_1, \cdots, \xi_{t-1})$-measurable. Given that $\mathbb{E}\left[\exp(Z_t^2/Y_t^2) \mid \xi_1, \cdots \xi_{t-1}\right] \leq \exp(1)$, for any $\lambda > 0$ and $\delta \in (0, 1)$ with probability at least $1 - \delta$,*

$$\sum_{t=1}^{T} Z_t \leq \frac{3}{4}\lambda \sum_{t=1}^{T} Y_t^2 + \frac{1}{\lambda}\log(1/\delta)$$

Next, we present a slightly modified version of the above lemma. Its proof follows the same lines with Lemma B.1 up to replacing $Y_t$ with a deterministic quantity, selecting a particular choice of $\lambda$ and dealing with the MDS $Z_t$ itself rather than its square, $Z_t^2$.

**Lemma B.2.** *Let $Z_1, \cdots, Z_T$ be a martingale difference sequence (MDS) with respect to random vectors $\xi_1, \cdots, \xi_T$ and $\sigma^2 \in \mathbb{R}$ such that $\mathbb{E}\left[\exp(Z_t/\sigma^2) \mid \xi_1, \cdots \xi_{t-1}\right] \leq 1$. Then, with probability as least $1 - \delta$,*

$$\sum_{t=1}^{T} Z_t \leq \sigma^2 \log(1/\delta)$$

We will also make use of another relevant result (Lemma 5 in Li & Orabona (2020)) regarding the probabilistic behavior of maximum over norms of noise vectors.

**Lemma B.3** (Lemma 5 in Li & Orabona (2020)). *Under assumptions as in Eq. (5) and (11), let $\xi_t = \nabla f(x_t, z_t) - \nabla f(x_t)$. For $\delta \in (0, 1)$, with probability at least $1 - \delta$,*

$$\max_{1 \leq t \leq T} \|\xi_t\|^2 \leq \sigma^2 \log\left(\frac{eT}{\delta}\right)$$

**Theorem 4.3.** *Let $x_t$ be generated by AdaGrad and define $\Delta_t = f(x_t) - \min_{x \in \mathbb{R}^d} f(x)$. Under sub-Gaussian noise assumption as in Eq. (11), with probability at least $1 - 3\delta$,*

$$\Delta_{t+1} \leq \Delta_1 + 3G_0^{-1}G^2 + 2G_0^{-1}\sigma^2 \log\left(\frac{et}{\delta}\right) + \frac{3}{4G_0}\sigma^2 \log(1/\delta)$$
$$+ \frac{L}{2}\left(1 + \log\left(\max\left\{1, G_0^2\right\} + 2G^2 t + 2\sigma^2 t \log\left(\frac{et}{\delta}\right)\right)\right)$$

*Proof.* Using the initial steps of the proof in the original derivation,

$$\Delta_{T+1} \leq \Delta_1 + \underbrace{\sum_{t=1}^{T} -\eta_t \|\bar{g}_t\|^2}_{(A)} + \underbrace{\sum_{t=1}^{T} -\eta_t \langle \bar{g}_t, \xi_t \rangle}_{(B)} + \underbrace{\frac{L}{2}\sum_{t=1}^{T} \eta_t^2 \|g_t\|^2}_{(C)}$$

**Term (A) + (B):** In order to deal with measurability issues, we will divide term (B) into two parts:

$$
\sum_{t=1}^{T} -\eta_t \langle \bar{g}_t, \xi_t \rangle = \sum_{t=1}^{T} -\eta_{t-1} \langle \bar{g}_t, \xi_t \rangle + \sum_{t=1}^{T} (\eta_{t-1} - \eta_t) \langle \bar{g}_t, \xi_t \rangle
$$

$$
\overset{(1)}{\leq} \sum_{t=1}^{T} -\eta_{t-1} \langle \bar{g}_t, \xi_t \rangle + 2(G^2 + \max_{1 \leq t \leq T} \|\xi_t\|^2) \sum_{t=1}^{T} (\eta_{t-1} - \eta_t)
$$

$$
\leq \sum_{t=1}^{T} -\eta_{t-1} \langle \bar{g}_t, \xi_t \rangle + 2(G^2 + \max_{1 \leq t \leq T} \|\xi_t\|^2) \eta_0
$$

where we used Cauchy-Schwarz together with Young's inequality to obtain inequality (1) and telescoped in the last line. Moreover, we pick $\eta_0 \geq \eta_1$ to make sure monotonicity. Without loss of generality, a natural choice would be $\eta_0 = G_0^{-1}$, which aligns with the definition in Algorithm 1. By Lemma B.3, with probability at least $1 - \delta$,

$$
\sum_{t=1}^{T} -\eta_t \langle \bar{g}_t, \xi_t \rangle \leq \sum_{t=1}^{T} -\eta_{t-1} \langle \bar{g}_t, \xi_t \rangle + 2G_0^{-1} \left( G^2 + \sigma^2 \log \left( \frac{eT}{\delta} \right) \right)
$$

Now, we will invoke Lemma B.1 on the term $\sum_{t=1}^{T} -\eta_{t-1} \langle \bar{g}_t, \xi_t \rangle$ by setting $Z_t = -\eta_{t-1} \langle \bar{g}_t, \xi_t \rangle$, $Y_t^2 = \eta_{t-1}^2 \|\bar{g}_t\|^2 \sigma^2$, with probability at least $1 - \delta$,

$$
\sum_{t=1}^{T} -\eta_{t-1} \langle \bar{g}_t, \xi_t \rangle \leq \frac{3}{4} \lambda \sum_{t=1}^{T} Y_t^2 + \frac{1}{\lambda} \log(1/\delta)
$$

$$
= \frac{3}{4} \lambda \sigma^2 \sum_{t=1}^{T} \eta_{t-1}^2 \|\bar{g}_t\|^2 + \frac{1}{\lambda} \log(1/\delta)
$$

Now, summing up the expression above with term (A) and leaving $\frac{1}{\lambda} \log(1/\delta)$ aside for now,

$$
\frac{3}{4} \lambda \sigma^2 \sum_{t=1}^{T} \eta_{t-1}^2 \|\bar{g}_t\|^2 - \sum_{t=1}^{T} \eta_t \|\bar{g}_t\|^2 \leq \frac{3}{4} \lambda \sigma^2 \sum_{t=1}^{T} \eta_{t-1}^2 \|\bar{g}_t\|^2 - G_0 \sum_{t=1}^{T} \eta_t^2 \|\bar{g}_t\|^2
$$

$$
\leq \frac{3}{4} \lambda \sigma^2 \sum_{t=1}^{T} \eta_{t-1}^2 \|\bar{g}_t\|^2 - G_0 \sum_{t=1}^{T} \eta_{t-1}^2 \|\bar{g}_t\|^2 + G_0 \sum_{t=1}^{T} \left( \eta_{t-1}^2 - \eta_t^2 \right) \|\bar{g}_t\|^2
$$

$$
\leq \left( \frac{3}{4} \lambda \sigma^2 - G_0 \right) \sum_{t=1}^{T} \eta_{t-1}^2 \|\bar{g}_t\|^2 + G_0 G^2 \eta_0^2
$$

where we used $G_0 \eta_t^2 \leq \eta_t$ in the first inequality and added/subtracted $\sum_{t=1}^{T} \eta_{t-1}^2 \|\bar{g}_t\|^2$ in the second inequality. Since we have a free variable to choose, $\lambda$, we could set it to $\lambda = \frac{4G_0}{3\sigma^2}$ to obtain,

$$
\frac{3}{4} \lambda \sigma^2 \sum_{t=1}^{T} \eta_{t-1}^2 \|\bar{g}_t\|^2 - \sum_{t=1}^{T} \eta_t \|\bar{g}_t\|^2 \leq G_0^{-1} G^2
$$

Hence, summing up all the expressions together, with probability at least $1 - 2\delta$,

$$
(A) + (B) \leq 3G_0^{-1} G^2 + 2G_0^{-1} \sigma^2 \log \left( \frac{eT}{\delta} \right) + \frac{3}{4G_0} \sigma^2 \log(1/\delta)
$$

**Term (C):** This term is easy to prove using online learning lemmas as we did previously, but we introduce a slight change in order to avoid bounded stochastic gradient assumption.

$$\frac{L}{2}\sum_{t=1}^{T}\eta_t^2\|g_t\|^2 \leq \frac{L}{2}\left(1+\log\left(\max\left\{1,G_0^2\right\}+\sum_{t=1}^{T}\|g_t\|^2\right)\right)$$

$$\leq \frac{L}{2}\left(1+\log\left(\max\left\{1,G_0^2\right\}+2\sum_{t=1}^{T}\|\bar{g}_t\|^2+2\sum_{t=1}^{T}\|\xi_t\|^2\right)\right)$$

$$\leq \frac{L}{2}\left(1+\log\left(\max\left\{1,G_0^2\right\}+2G^2T+2\left(\max_{1\leq t\leq T}\|\xi_t\|^2\right)T\right)\right)$$

We invoked Lemma 4.2 to obtain the first inequality. Once again via Lemma B.3, with probability at least $1-\delta$,

$$\frac{L}{2}\sum_{t=1}^{T}\eta_t^2\|g_t\|^2 \leq \frac{L}{2}\left(1+\log\left(\max\left\{1,G_0^2\right\}+2G^2T+2\sigma^2T\log\left(\frac{eT}{\delta}\right)\right)\right)$$

Finally, merging all the expression, with probability at least $1-3\delta$,

$$\Delta_{T+1} \leq \Delta_1 + 3G_0^{-1}G^2 + 2G_0^{-1}\sigma^2\log\left(\frac{eT}{\delta}\right) + \frac{3}{4G_0}\sigma^2\log(1/\delta)$$

$$+ \frac{L}{2}\left(1+\log\left(\max\left\{1,G_0^2\right\}+2G^2T+2\sigma^2T\log\left(\frac{eT}{\delta}\right)\right)\right)$$

$$= O\left(\Delta_1 + \sigma^2\log\left(\frac{eT}{\delta}\right) + \sigma^2\log(1/\delta) + L\log\left(T+\sigma^2T\log\left(\frac{eT}{\delta}\right)\right)\right)$$

$$\square$$

## B.2 HIGH PROBABILITY CONVERGENCE RATE

Now, we are at a position to prove noise-adaptive bounds.

**Theorem 4.4.** *Let $x_t$ be generated by AdaGrad and define $\Delta_t = f(x_t) - \min_{x\in\mathbb{R}^d} f(x)$. Under sub-Gaussian noise assumption as in Eq. (11) and considering high probability boundedness of $\Delta_{max}$ due to Theorem 4.3, with probability at least $1-5\delta$,*

$$\frac{1}{T}\sum_{t=1}^{T}\|\bar{g}_t\|^2 \leq \frac{32\left(\Delta_{max}+L\right)^2 + 8\left(\Delta_{max}+L\right)\left(G_0+\sigma\sqrt{2\log(1/\delta)}\right) + 8\sigma^2\log(1/\delta)}{T} + \frac{8\sqrt{2}\left(\Delta_{max}+L\right)\sigma}{\sqrt{T}}$$

*Proof.* We take off from the same step of the original analysis by defining $\xi_t = \nabla f(x_t, z_t) - \nabla f(x_t)$,

$$\sum_{t=1}^{T}\|\bar{g}_t\|^2 \leq \frac{\Delta_{\max}}{\eta_T} + \sum_{t=1}^{T}-\langle\bar{g}_t,\xi_t\rangle + \frac{L}{2}\sum_{t=1}^{T}\eta_t\|g_t\|^2$$

$$\leq (\Delta_{\max}+L)\sqrt{G_0^2+\sum_{t=1}^{T}\|g_t\|^2} + \sum_{t=1}^{T}-\langle\bar{g}_t,\xi_t\rangle$$

$$\leq (\Delta_{\max}+L)\sqrt{G_0^2+2\sum_{t=1}^{T}\left(\|\bar{g}_t\|^2+\|\xi_t\|^2+\sigma^2-\sigma^2\right)} + \sum_{t=1}^{T}-\langle\bar{g}_t,\xi_t\rangle$$

$$\leq (\Delta_{\max}+L)\left(G_0+\sqrt{2\sum_{t=1}^{T}\|\bar{g}_t\|^2+\left(\underbrace{\max\left\{0,2\sum_{t=1}^{T}(\|\xi_t\|^2-\sigma^2)\right\}}_{(*)}\right)^{1/2}}+\sigma\sqrt{2T}\right) + \underbrace{\sum_{t=1}^{T}-\langle\bar{g}_t,\xi_t\rangle}_{(**)}$$

We already showed that $-\langle \bar{g}_t, \xi_t \rangle$ is a MDS. Similarly, we could show that martingale property holds for $\|\xi_t\|^2 - \sigma^2$,

$$\mathbb{E}\left[\,\|\xi_t\|^2 - \sigma^2 \mid \xi_1, \cdots, \xi_{t-1}\,\right] = \mathbb{E}\left[\,\|\xi_t\|^2 \mid \xi_1, \cdots, \xi_{t-1}\,\right] - \sigma^2 \le \sigma^2 - \sigma^2 = 0.$$

Lemma B.2 immediately implies for term $(*)$ that with probability at least $1 - \delta$,

$$\sum_{t=1}^{T}(\|\xi_t\|^2 - \sigma^2) \le \sigma^2 \log(1/\delta)$$

For term $(**)$, we apply Lemma B.1 with $Y_t^2 = \sigma^2 \|\bar{g}_t\|^2$ and $\lambda = 1/\sigma^2$ to obtain with probability at least $1 - \delta$,

$$\sum_{t=1}^{T} -\langle \bar{g}_t, \xi_t \rangle \le \frac{3}{4}\sum_{t=1}^{T} \|\bar{g}_t\|^2 + \sigma^2 \log(1/\delta)$$

Plugging these values in and re-arranging,

$$\frac{1}{4}\sum_{t=1}^{T} \|\bar{g}_t\|^2 \le \sqrt{2}\,(\Delta_{\max} + L)\sqrt{\sum_{t=1}^{T} \|\bar{g}_t\|^2} + (\Delta_{\max} + L)\left(G_0 + \sigma\sqrt{2\log(1/\delta)} + \sigma\sqrt{2T}\right) + \sigma^2 \log(1/\delta)$$

We will conclude our proof by treating the above inequality as a quadratic inequality with respect to $x = \sqrt{\sum_{t=1}^{T} \|\bar{g}_t\|^2}$. Defining $c = (\Delta_{\max} + L)\left(G_0 + \sigma\sqrt{2\log(1/\delta)} + \sigma\sqrt{2T}\right) + \sigma^2 \log(1/\delta)$,

$$x^2 - 4\sqrt{2}\,(\Delta_{\max} + L)\,x - 4c \le 0,$$

where the roots of the inequality are

$$x = \frac{4\sqrt{2}\,(\Delta_{\max} + L) \pm \sqrt{32\,(\Delta_{\max} + L)^2 + 16\,(\Delta_{\max} + L)\left(G_0 + \sigma\sqrt{2\log(1/\delta)} + \sigma\sqrt{2T}\right) + 16\sigma^2 \log(1/\delta)}}{2}$$

Since $x > 0$ by default, we will take into account the positive root above, which yields,

$$\sum_{t=1}^{T} \|\bar{g}_t\|^2 \le 4\left(\sqrt{2}\,(\Delta_{\max} + L) + \sqrt{(\Delta_{\max} + L)\left(G_0 + 2\,(\Delta_{\max} + L) + \sigma\sqrt{2\log(1/\delta)} + \sigma\sqrt{2T}\right) + \sigma^2 \log(1/\delta)}\right)^2$$

$$\frac{1}{T}\sum_{t=1}^{T} \|\bar{g}_t\|^2 \le \frac{32\,(\Delta_{\max} + L)^2 + 8\,(\Delta_{\max} + L)\left(G_0 + \sigma\sqrt{2\log(1/\delta)}\right) + 8\sigma^2 \log(1/\delta)}{T} + \frac{8\sqrt{2}\,(\Delta_{\max} + L)\,\sigma}{\sqrt{T}}$$

$$\square$$

