# OpenReview forum: "High Probability Bounds for a Class of Nonconvex Algorithms with AdaGrad Stepsize"
_ICLR.cc/2022/Conference — ICLR 2022 Poster_

### Official Review · Reviewer_GqwP · 2021-10-23

**Correctness:** 4
**Technical Novelty And Significance:** 3
**Empirical Novelty And Significance:** 3
**Recommendation:** 8
**Confidence:** 3

**Main Review:**

I believe the conclusion is meaningful and important. The writing and presentation clearly shows the main steps of reasoning. By partitioning the proof into three steps in Sec. 4.2, with their corresponding lemmas, I can easily see the ideas and pinpoint to each separate proof.

I can barely find weaknesses of the paper. I have a few questions below.

1. As far as I remember, the AdaGrad usually means an entrywise step size. Compared to the paper, I would rather say $\eta_{i,t} = (G_0^2 + \sum g_{k,i}^2)^{-1/2}$ and $x_{t+1} = x_t - diag(\eta_{i,t}) g_t$. The diagonal matrix approximates the inversed Hessian to accelarate. Does this paper apply to that case, and will there be an acceleration if the Hessian is truly diagonal?

2. I'm not quite familiar with the high probability reasoning, so it could be helpful to add in appendix a comparison with the papers in "High probability results" paragraph, say, what algorithms did they use, what are their result.

3. I think the usual "convergence in expectation" result also indicates the probability. $f(x)-f(x^*)$ is lower bounded by $0$, so a small expectation $Ef(x)-f(x^*)<\epsilon$ means that "with probability $p$, $f(x)-f(x^*)<\delta(p)$". Does a standalone argument about probability benefit?

**Summary Of The Paper:**

This paper proves a high probability convergence rate for the first order algorithm, achieving the optimality of the probability and the convergence rate under this type of algorithm.

**Summary Of The Review:**

I think the contribution is meaningful and the writing is clear. If other reviewers are not doubtful about the novelty, then I'll give an accept.

---

> ### Author Response · Authors · 2021-11-15
> **Thank you for your review!**
>
> Thank you very much for your positive feedback and encouraging comments! We will respond to each of your comments individually.
>
> We present our analysis in a structured manner to showcase its versatility and explain the advantages it has with respect to standard analysis clearly, we thank you for recognizing our efforts of clarity.
>
> **1)** Your assessment is correct that AdaGrad originally has per-coordinate step-size. Could you please explain your following statement in more details *“Does this paper apply to that case, and will there be an acceleration if the Hessian is truly diagonal?”* We will appreciate it very much if you could explain what you mean by acceleration. If we understand your comment correctly, original AdaGrad paper proposes two variants of matrix-valued stepsize: a full matrix version that accumulates outer products of gradients and a diagonal matrix version that accumulates only the diagonal of the outer products. In practice, either the diagonal matrix version or the scalar step-size version (the one we consider) is used, because inverting a full matrix is too costly.
>
> It is a bit difficult to extend our analysis, in the same structure, to diagonal matrix version with per-coordinate step sizes due to the term $\sum_{t=1}^T-\langle\bar g_t,\xi_t\rangle $ for which positivity of any summands is not guaranteed. One perspective to tackle this setting is; first, bounding $\sum_{t=1}^T-\langle\bar g_t,\xi_t\rangle$ in a similar spirit as bounding $\Delta_t$ in Proposition 4.2. Eventually, by using some algebraic tricks, we could end up with an expression of the form $\sum_{t=1}^T\langle\eta_{t-1}\bar g_t,\bar g_t\rangle\leq B \sqrt{ \sum_{t=1}^T \langle\eta_{t-1} \bar g_t, \bar g_t\rangle}+C$, where B and C include absolute constants and $\log(T)$ terms. Finally, we might treat this as a quadratic inequality of the form $x^2 - Bx - C \leq 0$, with $X = \sqrt{\sum_{t=1}^T\langle\eta_{t-1}\bar g_t,\bar g_t\rangle}$, and solve for X as we do in the revised supplementary material. You could take a look at the derivation that follows the line *“We will conclude our proof by treating the above inequality...”* on page 27 for an intuition about this quadratic inequality representation. There are 2 possible downsides of this analysis:
> - Dependence on problem parameters might be slightly worse than before as we have to use loose bounds for “solving the quadratic inequality”.
> - We won’t have the same rate in the deterministic setting as in Theorem 4.1.
>
> Indeed, it is an interesting future direction to come up with a better analysis in the per-coordinate step-size setting. We believe that there should be a way to extend our approach to this setting.
>
> **2)** We could create a table in the appendix that displays existing high probability results for smooth nonconvex optimization and explain their advantages/disadvantages in a comparative fashion. Thank you for the suggestion!
>
> **3)** For the problem of finding first-order stationary points for general smooth minimization (which is the setting of our paper), the standard convergence metric is taking gradient norm to zero; specifically it is either $\min_{1 \leq t \leq T} \mathbb E [ \| \nabla f(x_t) \|^2] $ or $\frac{1}{T} \sum_{t=1}^{T} \mathbb E [ \| \nabla f(x_t) \|^2 ] $ [1, 2, 3, 4, 5, 6, 7]. Simply using Markov’s inequality, one can immediately show high probability results. However, this yields a dependence of $\frac{1}{\delta}$ on the probability margin ($\delta \in (0,1)$), which is worse than the best-known dependence of $\log \left( \frac{1}{\delta} \right)$.
>
> In order to achieve better dependence, we need exponential-type concentration inequalities such as Bernstein’s inequality (for random variables) or Freedman’s inequality (for martingale differences). These inequalities eventually lead to measurability problems where step-size $\eta_t$ and gradient norm $\| \nabla f(x_t)\|^2$ are dependent random variables. In essence, our proposed techniques introduce a way of solving this problem with complementary quantification of bounds over function value growth/behavior.
>
> **References:**
>
> [1] Saeed Ghadimi and Guanghui Lan. Stochastic first-and zeroth-order methods for nonconvex stochastic programming, 2013.
>
> [2] Rachel Ward, Xiaoxia Wu, and Leon Bottou. AdaGrad stepsizes: Sharp convergence over nonconvex landscapes, 2019.
>
> [3] Dongruo Zhou, Yiqi Tang, Ziyan Yang, Yuan Cao, and Quanquan Gu. On the convergence of adaptive gradient methods for nonconvex optimization, 2018.
>
> [4] Xiaoyu Li and Francesco Orabona. A high probability analysis of adaptive sgd with momentum, 2020.
>
> [5] Ashok Cutkosky and Harsh Mehta. High-probability bounds for non-convex stochastic opti- mization with heavy tails, 2021.
>
> [6] Yoel Drori and Ohad Shamir. The complexity of finding stationary points with stochastic gradient descent, 2021.
>
> [7] Zeyuan Allen-Zhu. How to make the gradients small stochastically: Even faster convex and nonconvex sgd, 2021.

---

> > ### Comment · Reviewer_GqwP · 2021-11-15
> > **Thanks for the feedback**
> >
> > I appreciate the detailed feedback. About my question "Does this paper apply to that case, and will there be an acceleration if the Hessian is truly diagonal", I think the first half is addressed. I was just wondering whether "per-coordinate step size" ends up with a better rate/bound than this paper's result, but it's not necessary. I would suggest the author(s) changing the title and some text that, actually this paper uses a "simplified" AdaGrad, not the typical AdaGrad with "per-coordinate step size", so that it avoids overclaiming and confusion. It can be named as something like "self-normalizing step size".
> >
> > If there is no new theory about the per-coordinate AdaGrad, I think I'll keep my assessment since I already gave a high score.

---

> > > ### Author Response · Authors · 2021-11-16
> > > **Thank you for the quick response**
> > >
> > > Please find our response to your further comments below.
> > >
> > > - *"I was just wondering whether "per-coordinate step size" ends up with a better rate/bound"*. The bound will be the same with respect to dependence on $T$ and $L$, and there won't be any non-asymptotic rate improvement. The main diffence would be the additional dependence on dimension. For instance, instead of $G$, we would have $G_\infty \sqrt{d}$. However, these quantities are more or less in the same order due to relationship between $\ell_2$ and $\ell_\infty$ norms. The difference in the final bounds will be minor and there won't be a speed up.
> > >
> > >
> > > - *"I would suggest the author(s) changing the title"*. As you mentioned, we didn't want to overclaim in the first place, hence we especially used the term "Nonconvex Algorithms *with AdaGrad stepsizes*" in the title. We also mention this a few times in the paper to eliminate confusion. In fact, in [1], the authors consider the same, scalar step-size version of AdaGrad which they denote as AdaGrad-Norm, and their title has the same expression, "AdaGrad stepsizes". We could add this clarification in the abstract as they do in their paper. We would appreciate your follow-up opinion on this.
> > >
> > >
> > > - We don't have a full theory for per-coordinate step-sizes but what we described in our first response would give us a rate of $O(1 / \sqrt{T})$, though it might be a challenge to obtain $O(1/T)$ rate in the deterministic setting. During the rebuttal period, we showed a new result for our setting under sub-Gaussian noise, which we mention in our response to reviewer BF1x. Our new result shows that AdaGrad with scalar step-size adapts to noise, and achieves the faster rate $O(1/T)$ as $\sigma \to 0$, while removing dependence on stochastic gradient bounds. We will appreciate it if you could take a look at our revised supplementary material, Appendix B for the full proof.
> > >
> > >
> > > **References:**
> > >
> > > [1] Rachel Ward, Xiaoxia Wu, and Leon Bottou. AdaGrad stepsizes: Sharp convergence over nonconvex landscapes, 2019.

---

### Official Review · Reviewer_BF1x · 2021-11-02

**Correctness:** 4
**Technical Novelty And Significance:** 3
**Empirical Novelty And Significance:** 3
**Recommendation:** 6
**Confidence:** 4

**Main Review:**

The paper addresses an interesting and relevant problem, and the difficulties present are significant.
I have two main concerns about the results currently.

1: Although it is true that the final result depends on the "unknown" $L$ and $\sigma$, it does not have a very good dependence on these quantities. In particular, the in-expectation results for adaptive methods would be more like
$$O\left(\sigma/\sqrt{T}\right)$$
while these results are more like $$O\left(\left(\sigma\sqrt{\log(1/\delta)}/\sqrt{T} + \tilde GL/\sqrt{T}\right)\right).$$ That is, when $\sigma\to 0$, the result does not have any asymptotic improvement. In fact, I believe a similar result is achievable with \emph{non-adaptive} SGD using learning rate $\eta_t = 1/\sqrt{t}$. In this case, applying standard martingale bounds to the second display of page 5 (which is fine since the learning rate is deterministic), would yield $$1/\sqrt{T} \sum_{t=1}^T \|\bar g_t\|^2 \le \Delta + \sqrt{\sigma^2 G^2 \log(T)\log(1/\delta)} + L\tilde G^2 \log(T),$$ from which some rearrangement provides a similar bound. Notably, SGD also doesn't require any knowledge of the smoothness or Lipschitz parameters. Thus, unless I miss some interpretation, this result seems to be more on the line of "the adaptive method doesn't *break* anything" rather than "the adaptive method is actually useful".

2: The high probability bound is a little weird, as it applies to just a sum of the gradients. What does this actually mean? For the in-expectation bounds I can actually produce an iterate that has a small gradient norm in expectation by randomly picking an iterate. Here if I randomly select an iterate I think the high-probability guarantee will be lost, so unless there is some other construction or better argument, it cannot actually provide any high-probability guarantees about any iterate. Since my feeling is that the point of the high-probability analysis is to say things about individual runs of the algorithm, this may limit the usefulness of the result. Now, this could be surmounted by simply calculating a batch stochastic gradient for each iterate as done in Ghadimi and Lan, but this seems to require a lot of gradient computations. Another alternative would be to use minibatch sizes of $\sqrt{T}$ for $\sqrt{T}$ iterations, but this would essentially average out all the variance and so we may possibly run into an even more extreme version of the issue in which the bounds do not obviously improve on non-adaptive methods.



**Summary Of The Paper:**

This paper provides an analysis of adaptive learning rate scheme in stochastic non-convex settings. Under assumptions of smoothness and bounded stochastic gradients (a light tailed condition), it is shown that the sum of the gradient norms of the iterates of the non-convex algorithm is small.
In order to accomplish this, the analysis must deal with the standard difficulty for adaptive learning rates, which is that the learning rate depends on the current iterate and so makes it harder to apply standard martingale inequalities.
The final results include a dependence on the smoothness constant $L$ and variance $\sigma$ which are not explicitly used in the algorithm.

**Summary Of The Review:**

The paper considers and interesting and difficult problem. However, the results seem to fall short of realizing the ideal goals.

---

> ### Author Response · Authors · 2021-11-14
> **Thank you for your review!**
>
> We would like to thank the reviewer for their detailed, constructive comments on our results and also the technical details! We appreciate your recognition of technical subtleties very much. We will respond to your points individually.
>
> 1- Your observation about noise adaptation (i.e., rate improvement to $1/T$ when $\sigma \to 0$) is very important and we regard it as a shortcoming to be improved in our paper, too. Indeed, we have a new, complementary result for noise adaptation that we want to share, which we presented in Appendix B in our revised supplementary pdf.
>
> First of all, none of the high probability results that we know of achieves noise adaptation under the same conditions (neither for SGD [1, 2, 3] nor for adaptive methods [4, 5]) without using the knowledge of variance. The only paper that we know to achieve noise adaptation is [6], which additionally assumes that noise behaves as sub-Gaussian while requiring the knowledge of Lipschitz constant.
>
> Under the same set of assumptions as [6], we managed to prove a new noise-adaptive high probability convergence rate of $O \left( \frac{(\Delta_{\text{max}} + L)^2 + (\Delta_{\text{max}} + L)(G_0 + \sigma \sqrt{\log (1 / \delta)}) + \sigma^2 \log(1 / \delta)}{T} + \frac{(\Delta_{\text{max}} + L) \sigma}{\sqrt{T}} \right)$ for AdaGrad without knowing L and $\sigma$. Additionally, this rate is independent of bounds on stochastic gradients.
>
> Although our rates in Theorem 4.3 and 5.1 for stochastic oracles are not noise-adaptive either, there is a fundamental difference; in Theorem 4.1 we prove that AdaGrad achieves $1/T$ convergence under deterministic gradients while SGD can only converge at $1 / \sqrt{T}$. Moreover, we would like to mention that SGD can converge without the knowledge of smoothness constant L if and only if the objective is additionally Lipschitz continuous. For general smooth functions, knowledge of L is essential together with $\eta_t = O(1 / \sqrt{t})$ to achieve convergence.
>
> Regarding your comments for SGD and adaptive methods, adaptive methods are known to outperform SGD variants for some machine learning tasks, e.g. attention models [7], while SGD has favorable qualities for other certain scenarios such as generalization performance [8]. There are different scenarios where one is more useful than the other.
>
> Our main goal in this paper is to prove complementary theoretical results to in-expectation counterparts with best-known dependence on parameters while being agnostic to their knowledge. We do so through a customized approach to overcome measurability problems while providing a simple and clear analysis. We believe our proof strategy will be useful for analysis of adaptive methods in general.
>
> 2- Our measure of convergence is standard among high probability results for smooth, non-convex problems [3, 4, 5, 6, 9, 10], which is also the standard for in-expectation analysis [11, 12], and we follow the same convention for consistency. Although, we agree with you on the fact that it is less informative than the last iterate convergence or with respect to a randomly selected iterate, this is the standard quantification for first-order stationarity convergence.
>
> As you pointed out, using a mini-batch may not be of help under the problem setting we consider and it will yield worse per-iteration cost which we need to handle in the analysis.
>
> We hope we are able to address your concerns in the best way possible. We will be delighted to engage in further discussions!
>
> **References:**
>
> [1] Guanghui Lan. An optimal method for stochastic composite optimization, 2012.
>
> [2] Saeed Ghadimi and Guanghui Lan. Accelerated gradient methods for nonconvex nonlinear and stochastic programming,
>  2016.
>
> [3] Saeed Ghadimi and Guanghui Lan. Stochastic first-and zeroth-order methods for nonconvex stochastic programming, 2013.
>
> [4] Rachel Ward, Xiaoxia Wu, and Leon Bottou. AdaGrad stepsizes: Sharp convergence over nonconvex landscapes, 2019.
>
> [5] Dongruo Zhou, Yiqi Tang, Ziyan Yang, Yuan Cao, and Quanquan Gu. On the convergence of adaptive gradient methods for nonconvex optimization, 2018.
>
> [6] Xiaoyu Li and Francesco Orabona. A high probability analysis of adaptive sgd with momentum, 2020.
>
> [7] Jingzhao Zhang, Sai Praneeth Karimireddy, Andreas Veit, Seungyeon Kim, Sashank J Reddi, Sanjiv Kumar, and Suvrit Sra. Why are adaptive methods good for attention models?, 2020.
>
> [8] Pan Zhou, Jiashi Feng, Chao Ma, Caiming Xiong, Steven C. H. Hoi, and E Weinan. Towards theoretically understanding why sgd generalizes better than adam in deep learning, 2019.
>
> [10] Ashok Cutkosky and Harsh Mehta. High-probability bounds for non-convex stochastic opti- mization with heavy tails, 2021.
>
> [11] Yoel Drori and Ohad Shamir. The complexity of finding stationary points with stochastic gradient descent, 2021.
>
> [12] Zeyuan Allen-Zhu. How to make the gradients small stochastically: Even faster convex and nonconvex sgd, 2021.

---

> > ### Comment · Reviewer_BF1x · 2021-11-29
> > **adaptivity improvement**
> >
> > I appreciate the improved adaptivity and will increase my score.
> >
> > As far as my second point, I am not so convinced. First, I’m not sure I agree with your list of references: Ghadimi and Lan actually do output a particular iterate by spending more oracle queries, and in fact Cutkosky and Mehta provide a way to output a particular iterate with small gradient norm *without* spending more oracle queries, and also actually do provide some kind of bound on the last iterate.
> >
> > Nevertheless I do agree that some other references do think in terms of simply bounding the sum. However, just because people have done this in the recent past doesn’t actually mean that it is an extremely reasonable thing to do - it seems more a failure of previous works that we don’t really need to be propagating.

---

### Official Review · Reviewer_goz1 · 2021-11-14

**Correctness:** 4
**Technical Novelty And Significance:** 2
**Empirical Novelty And Significance:** 2
**Recommendation:** 6
**Confidence:** 4

**Main Review:**

Disclaimer: My apologies for putting this review very late, due to I have been assigned as an emergency reviewer about two days ago.

strength:

- The high probability bound results provide more information than the expected convergence results (basically the variance of the optimization behaviours), while the AdaGrad method is a base for many widely used optimizers in machine learning, and therefore the topic is important to optimization community.
- The analysis is simple and elegant, which I appreciate. The proofs are correct as far as I have checked.
- The presentation is very clear and easy to follow.

weakness:

- I am concerned with the results do not tell us much information about why the AdaGrad and its variants enjoy good performance in practice. In particular,

- The main results in Theorem 4.2 claims that after $T$ iterations, the averaged cumulated gradient norm is small with high probability. However, it seems this result cannot exclude a bad case of saddle points, where the gradient norm is small, while since the normalizer in Algorithm 1 is the sum of the historical gradient norm (let's say $g_1$ has its norm at $\Theta(1)$), the AdaGrad would get stuck around the saddle points, since it will do very small updates around saddle points.

- A more desirable result to explain the effectiveness of AdaGrad would be to show that AdaGrad not only converges to stationary points (which is shown in this paper) but also converges to points with good second order information (local minima rather than saddle points).

- I found the sentence above Eq. (1) "by choosing to output ... uniformly at random from the set of query points ... is bounded" a bit confusing. Does this mean the last iteration is not guaranteed to have small gradient norm (I can imagine somehow the algorithm could have oscillation behaviour, but not sure if this is the case here)? And does this imply we need to do some post-processing after running the AdaGrad method (which to my knowledge is not the case)? If this can be turned into a more direct result of last iteration convergence result with high probability, that would be better, since as I am aware of, there are expectation convergence results for last iteration convergence.

minor: Lemma 4.2: $a_i$ in the denominator should be $a_k$?

**Summary Of The Paper:**

This paper proposed a new analysis for AdaGrad method in smooth and non-convex optimization, to get high probability convergence toward stationary points.

Based on some assumptions (Eqs. (2), (6) and (7)), i.e., Lipschitz, bounded variance of gradient estimates, and bounded stochastic gradient, the authors analyzed Algorithm 1 (AdaGrad), which does not use any information of the quantities in the assumptions. They show that (in Theorem 4.2), with high probability, the averaged cumulative gradient norm square is converging to $0$ at order of $\tilde{O}(\log{(1/\delta)}/\sqrt{T})$. The main difficulty is to derive high probability bounds for the two quantities in Eq. (10), i.e., Propositions 4.1 and 4.2.

The authors then generalized their analysis to the generic AGD template (Algorithm 2), which recovers AdaGrad, AdaGrad with averaging, adaptive RSAG, and AcceleGrad as special cases by choosing different $\alpha_t$, $\eta_t$ and $\gamma_t$ parameters. Similar high probability convergence results can be obtained for AdaGrad with averaging, and adaptive RSAG, but not for AcceleGrad.



**Summary Of The Review:**

Overall, I think this work provides a neat and elegant analysis for AdaGrad and several variants. It shows that AdaGrad behaves similarly in terms of expectation convergence and with high probability convergence. However, I found the results still unsatisfactory in terms of I did not see how those results could explain the good performances achieved by AdaGrad and its variants. To my understanding, this is more like verifying and improving existing analysis to some extent but lack of providing new insights.

---

> ### Author Response · Authors · 2021-11-16
> **Thank you very much for your review!**
>
> We would like to thank you very much for your efforts in providing a review in such a short amount of time! Please find our responses below.
>
> **1)** *“the AdaGrad would get stuck around the saddle points”*. We don’t know of any theoretical result on AdaGrad and saddle avoidance. AdaGrad step-size will decrease very slowly as it approaches a saddle and it won’t decrease at fixed rate as SGD, maintaining a larger step-size than SGD. Also, [1] shows that SGD with injected isotropic noise avoids saddles, which might also apply to AdaGrad.
>
> **2)** Regarding your comment on second-order stationarity, it is a different line of research that demands different set of assumptions and analysis. The most relevant direction would be non-asymptotic high probability convergence to second-order stationary points. To our knowledge, there are no results for AdaGrad in this direction.
>
> First, [1,2] quantify iteration complexity of SGD to reach a second-order stationary point. To guarantee that, they need to assume Lipschitz-continuous gradients and Hessians, with additional assumptions on the noise model, which is more restrictive than our assumption set. [1] also assumes bounded objective suboptimality, for which we show a high probability bound on. In order to operate, they require knowledge of Lipschitz constants for gradient and Hessian, a bound on stochastic gradients, properties of noise [2] or curvature [1].
>
> For adaptive methods, [3] provides high probability convergence rates for RMSProp, but they assume *Hessian Lipschitzness* and additional assumption on stochastic gradients, which is more restrictive than our setting. Besides, their algorithm requires knowledge of Lipschitz constants for both gradient and Hessian.
>
> Convergence of AdaGrad to second-order stationary points is an interesting open problem and we totally agree with you on the value of such a result, however, this is a different line of research and it is beyond the scope this paper.
>
> **3)** In general, second-order convergence requires additional, restrictive assumptions and complex parameter choices that depend on problem constants which are difficult to even estimate in practice. In retrospect, our results are practically-viable and don't require knowledge of properties of loss, data or noisy nature of gradients. We specifically focus on vastly-recognized problem of *non-asymptotic, first-order convergence for smooth functions*. In this setting, ours is the first adaptive high probability analysis achieving best-known $\delta$ dependence while being agnostic to $L$, $\sigma$ and $G$, simultaneously. We also provide a new, improved analysis in Appendix B, which validates noise-adaptation property of AdaGrad under sub-Gaussian noise. With the new result, we unify best-known results in different fronts [4,5,6] under a single analysis. We believe this is a key contribution and our technique will open up new fronts in the theory of adaptive methods.
>
> To clarify, both for in-expectation and high probability analysis for first-order stationarity, $\frac{1}{T}\sum_{t=1}^T\mathbb E\|\nabla f(x_t)\|^2$ is a standard convergence quantification metric [3,4,5,6,7,8], and we follow the same standard for consistency.
>
> *"I found the sentence above Eq. (1) “...” a bit confusing"*. We refer to a known trick for in-expectation analysis. Defining $\hat x$ by uniformly randomly choosing among {$x_1,…,x_T$}, we could write $\frac{1}{T}\sum_{t=1}^T\|\nabla f(x_t)\|^2=\mathbb E[\| \nabla f(\hat x)\|^2]$. There is an equivalent trick for high probability analysis [7], which requires a two-stage algorithm. Since we focus on AdaGrad as is, we don’t apply this construction to our case. Also, AdaGrad iterates don’t need post-processing.
>
> *"there are expectation convergence results for last iteration convergence"*. Could you please point us to a specific result so that we could answer your concerns in a more concrete way? It would be very interesting to have in expectation convergence in last iterate for first-order stationarity.
>
> We hope we managed to respond to your comments thoroughly. We are always open for further discussions and clarifications!
>
> **References**
>
> [1] Rong Ge et al. Escaping from saddle points — online stochastic gradient for tensor decomposition, 2015.
>
> [2] Hadi Daneshmand et al. Escaping saddles with stochastic gradients, 2018.
>
> [3] Matthew Staib et al. Escaping saddle points with adaptive gradient methods, 2019.
>
> [4] Rachel Ward et al. AdaGrad stepsizes: Sharp convergence over nonconvex landscapes, 2019.
>
> [5] Xiaoyu Li and Francesco Orabona. A high probability analysis of adaptive sgd with momentum, 2020.
>
> [6] Dongruo Zhou et al. On the convergence of adaptive gradient methods for nonconvex optimization, 2018.
>
> [7] Saeed Ghadimi and Guanghui Lan. Accelerated gradient methods for nonconvex nonlinear and stochastic programming, 2016.
>
> [8] Yoel Drori and Ohad Shamir. The complexity of finding stationary points with stochastic gradient descent, 2021.

---

> > ### Comment · Reviewer_goz1 · 2021-11-30
> > **Thanks for the response**
> >
> > I would like to thank the authors for the feedback. My main concerns are addressed. And I would increase my score to a 6.
> >
> > Sorry for not posting the reference. I was referring to the following work:
> >
> > Allen-Zhu, Natasha 2: Faster Non-Convex Optimization Than SGD.
> >
> > Best,

---

### Decision · Program_Chairs · 2022-01-20

**Decision:**

Accept (Poster)

**Comment:**

The paper provides a high probability analysis for Adagrad for smooth non-convex optimization and shows its rate of convergence to critical points. Both rates for deterministic optimization and for stochastic optimization are provided. The main contribution of the paper is that unlike for SGD they don’t require knowledge of smoothness parameter in advance and second, they prove high probability results.

The reviewers lean positively towards the paper. One of the reviewers comments about the comparison with SGD which has some merit. The main comparison of this paper is w.r.t. ward et al 2019 and Zhou et al 2018 both of which prove high probability results. However, both these works require prior knowledge of smoothness parameter. The other axis of comparison is w.r.t algorithms like spider by Fang et al 2018 which uses variance reduction type techniques to obtain the optimal rate for critical point (here it is 1/sqrt{T} for norm square which is T^{-1/4} for norm and spider is T^{-1/3} for norm). Of course, an argument can be made for the fact that the algorithm here is closer to what is used in practice and more importantly, the assumptions there are somewhat stronger.

In any case, the paper still has interesting results and I am leaning towards an accept.